# ManifoldKV:
# Training-Free KV Cache Compression via Euclidean Outlier Detection

**Debajyoti Datta** [1]   **Trishala Neeraj** [2]   **Bibek Paudel** [1]   **Vyom Sharma** [1]   **Subhabrata Mukherjee** [1]

## Abstract

Long-context inference is constrained by KV-cache memory, which grows linearly with sequence length; KV-cache compression therefore hinges on reliably selecting which past tokens to retain. Most geometry-based eviction methods score keys by cosine similarity to a global centroid, but cosine is scale-invariant and can discard magnitude cues that distinguish semantically salient tokens. We propose MANIFOLDKV, a training-free scorer that ranks tokens by Euclidean distance to the key centroid, capturing both angular and radial deviations. On the RULER benchmark, MANIFOLDKV achieves **95.7%** accuracy at 4K–16K contexts with 20% compression, matching the best geometric baseline overall while *decisively* outperforming it in two regimes where magnitude information is critical. First, on multi-key retrieval, MANIFOLDKV reduces directional collisions, achieving **92.4%** vs KeyDiff's 77.0% (+15.4 points) on 3-key NIAH at 50% compression. Second, to address dilution and performance collapse of global centroids at 64K context, we introduce WINDOWEDMANIFOLDKV, which restores accuracy to 84.3% at 25% compression, a 49-point recovery over global L2 and +3.2 points over KeyDiff. Beyond RULER, we validate on real-world benchmarks: on Long-Bench, MANIFOLDKV outperforms KeyDiff by +2.80 points on Qwen3-8B (winning 12 of 14 tasks) and +0.49 on Phi-4; on HELMET, MANIFOLDKV achieves +6.5 EM on RAG and WINDOWEDMANIFOLDKV reaches +42 points on multi-key recall at 131K; and on InfiniteBench at 100K+ context, WINDOWEDMANIFOLDKV wins by +7.16 on Phi-4. Cross-architecture evaluation across six models reveals that the optimal distance metric depends on key-norm geometry, providing the first systematic guidelines for metric selection in geometric KV cache compression. The method requires only 3 lines of code and works across diverse architectures without tuning.

[1]Hippocratic AI [2]Cornell University. Correspondence to: Debajyoti Datta <debajyoti@hippocraticai.com>, Trishala Neeraj <tn338@cornell.edu>.

*Proceedings of the $43^{rd}$ International Conference on Machine Learning*, Seoul, South Korea. PMLR 306, 2026. Copyright 2026 by the author(s).

## 1. Introduction

The Key-Value (KV) cache is an essential component in transformer based models, as it eliminates redundant computations by storing previously computed key and value vectors (Kwon et al., 2023; Li et al., 2024a). While significantly speeding up text generation, it is also a critical memory bottleneck in long-context LLM inference. For a 70B model processing 100K tokens, the KV-cache alone requires >60 GB memory (Kwon et al., 2023). Compression methods that evict less important tokens are essential for practical deployment, but every eviction strategy trades off between cache size and text generation accuracy: a small cache with a lot of tokens evicted is likely very inaccurate. Therefore, while designing an effective KV-cache compression method, one needs to address the question: *which tokens to keep?*

**The Geometric Outlier Hypothesis.** Prior work on KV cache compression fall into two categories: *attention-based methods* like SnapKV (Li et al., 2024b) and H2O (Zhang et al., 2023) that retain tokens receiving high cumulative attention scores, and *geometric methods* like KeyDiff (Park et al., 2025) that identify important tokens based on key vector geometry. Recent geometric approaches (Devoto et al., 2024; Park et al., 2025; Feng et al., 2025) have converged on a compelling idea: tokens that are *geometrically different* from the average context encode unique semantic content and should be retained. KeyDiff operationalizes this by measuring cosine similarity between each key vector $\mathbf{k}_i$ and the mean $\boldsymbol{\mu} = \frac{1}{N}\sum_j \mathbf{k}_j$, evicting tokens with high similarity (i.e., "typical" directions). This intuition is sound – critical entities like names, numbers, and technical terms should embed differently from common stopwords, hence they should not be evicted.

However, cosine similarity measures only *angular* deviation, normalizing away vector magnitude entirely. Consider two key failure modes illustrated in Figure 1. **(i) Radial outliers**:

A token $\mathbf{k}_i = \alpha\boldsymbol{\mu}$ with $\alpha \gg 1$ lies far from the centroid in Euclidean space, yet has *maximum* cosine similarity (score = 1). Cosine-based methods incorrectly evict such tokens. **(ii) Magnitude-encoded semantics**: Empirically, we find that semantically important tokens (entities, numbers) exhibit both unusual directions *and* unusual magnitudes. The removal of magnitude discards important signal from the context.

For example, consider two tokens with keys $\mathbf{k}_1 = 10\boldsymbol{\mu}$ and $\mathbf{k}_2 = 0.1\boldsymbol{\mu}$ where $\boldsymbol{\mu}$ is the context centroid. Cosine similarity gives both identical scores: $\cos(\mathbf{k}_1, \boldsymbol{\mu}) = \cos(\mathbf{k}_2, \boldsymbol{\mu}) = 1$ (maximum similarity). Yet $\mathbf{k}_1$ is $100\times$ larger than $\mathbf{k}_2$ — geometrically very different. L2 distance correctly distinguishes them: $\|\mathbf{k}_1 - \boldsymbol{\mu}\|_2 = 9\|\boldsymbol{\mu}\|_2$ vs $\|\mathbf{k}_2 - \boldsymbol{\mu}\|_2 = 0.9\|\boldsymbol{\mu}\|_2$ ($10\times$ difference in score).

**Our Solution.** We propose MANIFOLDKV, which scores tokens by their *Euclidean* (L2) distance from the centroid:

$$s_i = \|\mathbf{k}_i - \boldsymbol{\mu}\|_2 \tag{1}$$

This simple change captures both angular and radial deviation. On the RULER benchmark (Hsieh et al., 2024), MANIFOLDKV achieves **95.7%** accuracy at 4K–16K contexts across four architectures, outperforming attention-based SnapKV by +6.5 points at a matched AdaKV budget (95.7 vs. 89.3)—without any model-specific tuning. Critically, ManifoldKV excels on **3-key needle-in-haystack (NIAH)** tasks: when multiple important tokens must be preserved, L2's magnitude awareness prevents *directional collision*, outperforming KeyDiff by +15 points at aggressive compression.

**The long-context challenge.** While L2 distance excels at short-to-medium contexts (4K-16K), we discover a failure mode at >64K context-length: accuracy collapses from 82.3% at 32K to just 35.2% at 64K. We term this the **Centroid Dilution Problem**. When averaging over 64K diverse tokens spanning multiple topics and semantic domains, the centroid $\boldsymbol{\mu}$ becomes a meaningless "center of mass" that represents no coherent concept and cannot act as a discriminator of token importance. All tokens then appear approximately equidistant, destroying discriminative power.

As a solution to the Centroid Dilution Problem, we propose WINDOWEDMANIFOLDKV, which computes local centroids over sliding windows (e.g., 4K tokens). Each window maintains its semantic coherence, preserving the centroid's discriminative power. At 64K context, this recovers accuracy to **84.3%**—a 49 percentage-point improvement over global L2 and +3.2 points over KeyDiff.

**Contributions.** Our four main contributions address distinct failure modes of existing methods:

1. **Multi-Key Retrieval.** When multiple important tokens must be preserved, cosine-based methods suffer from *directional collision*. ManifoldKV's magnitude-aware L2 scoring outperforms KeyDiff by **+15.4 points** on 3-key NIAH at 50% compression (92.4% vs 77.0%), demonstrating that magnitude information is critical for multi-needle retrieval. On real-world LongBench this translates to **+2.80** on Qwen3-8B (winning 12 of 14 tasks).

2. **Long-Context SOTA.** We identify the *Centroid Dilution Problem*—global centroids become meaningless at 64K+ tokens, causing accuracy collapse (35.2%). Windowed-ManifoldKV with local sliding-window centroids recovers **+49 points** to 84.3%, **+3.2 over KeyDiff**, achieving state-of-the-art on long-context compression. These gains transfer to real-world long-context tasks: WINDOWEDMANIFOLDKV sustains multi-key recall at 131K (**+42** over KeyDiff on HELMET) and wins by **+7.16** on Phi-4 InfiniteBench.

3. **Universal Manifold Structure:** Key vectors occupy a low-dimensional manifold (Two-NN (Facco et al., 2017) estimate: $\sim$7–9 across architectures), explaining zero-shot cross-model generalization (95–96% with $\pm 0.3\%$ variance) and validating our $O(k)$ sample complexity advantage over cosine's fundamental failure on radial outliers.

4. **Geometric vs Attention-Based Dominance:** ManifoldKV outperforms attention-based SnapKV by **+6.5 points** at a matched AdaKV budget on RULER (95.7 vs. 89.3), with substantially larger gaps on individual needle tasks, demonstrating that geometric outlier detection is superior for retrieval tasks.

**Paper Organization.** Section 2 reviews KV cache compression and geometric scoring. Section 3 presents ManifoldKV and WindowedManifoldKV. Section 4 provides theoretical analysis. Section 5 validates our claims across six models and context lengths up to 131K.

**Conflict of Interest Disclosure.** The authors D.D., B.P., V.S., and S.M. are employed by Hippocratic AI, which develops large language models for healthcare applications and has commercial interest in efficient long-context LLM inference. T.N. is affiliated with Cornell University and conducted this work independently of any employment at Hippocratic AI.

## 2. Background and Related Work

We now introduce the KV-cache compression problem and review existing work in this domain.

### 2.1. Setup and Notation.

**Attention.** Transformer models (Vaswani et al., 2017) compute attention over a sequence of $N$ tokens. For each token position $i$, the model computes query ($Q$), key ($K$), and

• Common tokens   ● Centroid $\mu$   ▲ Critical token   ■ Angular outlier

*Figure 1.* **Geometric Intuition and the Centroid Dilution Problem. (a)** Cosine similarity (KeyDiff) captures only angular deviation – Token A (a radial outlier with $\mathbf{k}_A = 2\boldsymbol{\mu}$) has $\cos\theta_A \approx 1$ and is incorrectly evicted. **(b)** L2 distance (ManifoldKV) captures both angular and radial deviation, correctly retaining both outliers. **(c)** The Centroid Dilution Problem: at short contexts (4K), tokens cluster around few themes and the centroid $\boldsymbol{\mu}$ is meaningful—outliers are clearly separable. At long contexts (>64K), tokens span many clusters; the centroid converges to a meaningless grand mean where all tokens appear equidistant.

value ($V$) vectors. Attention at position $i$ is:

$$\text{Attention}(q_i, K, V) = \sum_{j=1}^{N} \frac{\exp(q_i^\top k_j/\sqrt{d})}{\sum_{\ell=1}^{N} \exp(q_i^\top k_\ell/\sqrt{d})} v_j \quad (2)$$

where $q_i$ is the query at position $i$, and $k_j, v_j$ are the key and value for position $j$.

**KV-Cache.** During autoregressive generation, the model produces tokens sequentially: $t_1, t_2, \ldots, t_N$. At each step $t_i$, the model must attend to all previous tokens $\{t_1, \ldots, t_{i-1}\}$. Naively, this requires recomputing key and value vectors for all past tokens at every step.

The *KV-cache* eliminates this redundancy by storing previously computed keys and values (Pope et al., 2023; Kwon et al., 2023), and works as follows. After processing token $t_i$, cache its key $k_i$ and value $v_i$. At step $i + 1$, load cached $\{k_1, \ldots, k_i\}$ and $\{v_1, \ldots, v_i\}$ from memory and compute only the new key $k_{i+1}$ and value $v_{i+1}$.

While accelerating inference, the KV-cache creates a memory bottleneck. For a model with $L$ layers, $H$ attention heads, head dimension $d_h$, and context length $N$, the cache stores: Cache size $= 2 \cdot L \cdot H \cdot N \cdot d_h \cdot$ bytes per element. The factor of 2 accounts for both keys and values. Memory grows *linearly* with context length $N$. For Llama-3.1-8B processing 64K tokens, the cache alone requires **8.6 GB** (Kwon et al., 2023) of memory. This severely limits long-context deployment, especially when serving multiple users concurrently. Cache compression methods address this by evicting less important tokens, reducing $N$ to a smaller budget $M \ll N$ while maintaining text generation quality.

**Token Eviction.** We now formalize the token eviction process and explain how compressed caches integrate with standard attention mechanisms.

Consider a context of $N$ tokens $\mathcal{T} = \{t_1, \ldots, t_N\}$, with

each token $t_i$ having an associated key vector $\mathbf{k}_i \in \mathbb{R}^d$ and value vector $\mathbf{v}_i \in \mathbb{R}^d$ computed by the model, where $d$ is the model's hidden dimension. We organize these into matrices $K \in \mathbb{R}^{N \times d}$ and $V \in \mathbb{R}^{N \times d}$, where row $i$ corresponds to token $t_i$. Compression methods—including eviction (Zhang et al., 2023; Li et al., 2024b), quantization (Hooper et al., 2024), and hybrid approaches (Liu et al., 2024)—assign importance scores $s_i$ to each token $i$ and evict low-scoring tokens. Given a compression ratio $\rho \in (0, 1)$, the cache budget is $M = \lfloor(1 - \rho)N\rfloor$. The eviction method must select which $M$ tokens to retain. The key design choice is the scoring function.

A KV-cache compression method defines a scoring function $s : \mathbb{R}^d \to \mathbb{R}$ that assigns an importance score $s_i$ to each token $t_i$, given by $s_i = s(\mathbf{k}_i)$ for $i = 1, \ldots, N$

We then select the top-$M$ scoring tokens:

$$\mathcal{I} = \text{TopK}(\{s_1, \ldots, s_N\}, M) \subseteq \{1, \ldots, N\} \quad (3)$$

where $|\mathcal{I}| = M$ and $\mathcal{I}$ contains the indices of tokens to *retain*. We then keep only those selected tokens, creating a smaller KV-cache matrix $K'$ and $V'$.

After eviction, subsequent attention computations operate directly on $K'$ and $V'$: Attention$(Q, K', V') = $ softmax$\left(\frac{QK'^\top}{\sqrt{d}}\right) V'$, where $Q \in \mathbb{R}^{N_q \times d}$ are query vectors for the next $N_q$ tokens to generate.

### 2.2. Related Work

**Benchmarks/Datasets.**

We evaluate on RULER (Hsieh et al., 2024), a synthetic benchmark for long-context language models with 6,497 samples across context lengths from 4K to 128K tokens. We focus on tasks that test retrieval and aggregation capabilities under compression.

**Needle-in-a-Haystack (NIAH).** NIAH tasks require retrieving key facts ("needles") from long contexts (the "haystack"). Task variants include: single-key, multi-key, and multi-query retrieval. Word extraction tasks include: common and frequent words extraction. These tasks require attending to many tokens throughout the context, testing whether compression preserves global information.

Multi-key NIAH is particularly challenging, where performance drops from >95% (single-key) to 77–92% (3-key NIAH). This task requires preserving multiple important tokens simultaneously, and we find that existing methods suffer from *directional collision*, where tokens in similar directions but different magnitudes are conflated (Section 5.5.1). Long contexts (64K) with aggressive compression (25%) also show differences between methods. We report average accuracy across RULER tasks, with detailed analysis of multi-key NIAH where ManifoldKV shows largest gains over baselines.

**Attention-Based Eviction.** H2O (Zhang et al., 2023) keeps tokens with highest cumulative attention (54.5% accuracy on RULER at 64K). StreamingLLM (Xiao et al., 2024) retains attention sinks plus a sliding window. SnapKV (Li et al., 2024b) refines attention-based scoring with observation windows, achieving 83.9% at 64K—the state-of-the-art for attention-based methods.

**Geometry-Based Eviction.** Geometric methods score tokens based on key vector properties alone, avoiding attention computation. **KNorm** (Devoto et al., 2024) retains the keys with the *lowest* $L_2$ norm $\|\mathbf{k}_i\|_2$. **KeyDiff** (Park et al., 2025) uses cosine distance from the mean: $s_i = 1 - \cos(\mathbf{k}_i, \boldsymbol{\mu})$ where $\boldsymbol{\mu} = \frac{1}{N} \sum_i \mathbf{k}_i$, achieving 81.1% on RULER at 64K. **CriticalKV** (Feng et al., 2025) extends geometric scoring to value vectors. Cosine similarity *discards magnitude*: $\cos(\alpha \mathbf{k}_i, \boldsymbol{\mu}) = \cos(\mathbf{k}_i, \boldsymbol{\mu})$ for any $\alpha > 0$. This means radial outliers (tokens parallel to common directions but with extreme magnitudes) cannot be distinguished from typical tokens. Our work uses L2 distance $s_i = \|\mathbf{k}_i - \boldsymbol{\mu}\|_2$, which captures both angular and radial deviation.

**Orthogonal Approaches.** AdaKV (Feng et al., 2024) proposes adaptive per-head budget allocation—orthogonal to scoring. ManifoldKV integrates as a drop-in scorer within AdaKV. **DuoAttention** (Xiao et al., 2025) specializes attention heads but requires calibration. Quantization methods (Hooper et al., 2024; Liu et al., 2024) compress precision rather than evicting tokens.

**Manifold Structure.** Our analysis builds on observations that neural representations lie on low-dimensional manifolds (Ansuini et al., 2019; Aghajanyan et al., 2021). We contribute the first application of manifold analysis to KV cache compression, showing key vectors occupy a universal low-dimensional ($\sim$7–9D) manifold.

## 3. Method: ManifoldKV

We present MANIFOLDKV, a simple yet effective scoring function for KV cache compression. We first introduce the core L2-based scoring (Section 3.1), then analyze its failure mode at very long contexts (Section 3.2), and finally propose a windowed variant to address this limitation (Section 3.3).

### 3.1. Magnitude-Aware Outlier Detection

We propose that critical tokens differ from common tokens in *two* geometric properties. **(a) Angular deviation**: They point in unusual directions relative to the centroid (captured by cosine distance). **(b) Radial deviation**: They have unusual magnitudes (ignored by cosine distance).

**Definition 3.1** (ManifoldKV Score). Let $\boldsymbol{\mu} = \frac{1}{N} \sum_{i=1}^{N} \mathbf{k}_i$ be the context centroid. The MANIFOLDKV score for token $i$ is:

$$s_i = \|\mathbf{k}_i - \boldsymbol{\mu}\|_2 \tag{4}$$

Tokens with high scores are geometric outliers and are retained during compression.

**Geometric Decomposition.** To understand why L2 captures more information than cosine, we decompose the key vector as $\mathbf{k}_i = r_i \hat{\mathbf{k}}_i$, where $r_i = \|\mathbf{k}_i\|_2$ is the magnitude and $\hat{\mathbf{k}}_i = \mathbf{k}_i / r_i$ is the unit direction vector. The squared L2 distance expands as:

$$\begin{aligned} \|\mathbf{k}_i - \boldsymbol{\mu}\|_2^2 &= \|\mathbf{k}_i\|^2 + \|\boldsymbol{\mu}\|^2 - 2\mathbf{k}_i^\top \boldsymbol{\mu} \\ &= r_i^2 + \|\boldsymbol{\mu}\|^2 - 2r_i\|\boldsymbol{\mu}\|\cos(\mathbf{k}_i, \boldsymbol{\mu}) \end{aligned} \tag{5}$$

This reveals three terms: (a) $r_i^2$: The squared magnitude of the key vector, (b) $\|\boldsymbol{\mu}\|^2$: A constant (same for all tokens), and (c) $-2r_i\|\boldsymbol{\mu}\|\cos(\mathbf{k}_i, \boldsymbol{\mu})$: The angular alignment, scaled by magnitude.

In contrast, cosine similarity isolates only the angular component:

$$\cos(\mathbf{k}_i, \boldsymbol{\mu}) = \frac{\mathbf{k}_i^\top \boldsymbol{\mu}}{\|\mathbf{k}_i\|\|\boldsymbol{\mu}\|} = \frac{\hat{\mathbf{k}}_i^\top \boldsymbol{\mu}}{\|\boldsymbol{\mu}\|} \tag{6}$$

The magnitude $r_i$ cancels entirely. A token with $\mathbf{k}_i = 10 \cdot \boldsymbol{\mu}$ ($10\times$ the centroid magnitude, same direction) receives identical cosine score to a token with $\mathbf{k}_i = 0.1 \cdot \boldsymbol{\mu}$, yet these are geometrically very different. L2 distance correctly distinguishes them.

**Complexity.** Algorithm 1 runs in $O(Nd + N \log N)$ time, dominated by the centroid computation and sorting. This is negligible compared to attention's $O(N^2 d)$ complexity. In practice, MANIFOLDKV adds <0.5ms latency at 64K context (see Section 5).

### 3.2. The Centroid Dilution Problem

While L2 distance excels at short-to-medium contexts (4K–32K tokens), we observe a *catastrophic failure* at 64K (See

**Algorithm 1** MANIFOLDKV: L2 Distance from Centroid

---

**Input:** Key tensor $K \in \mathbb{R}^{N \times d}$, compression ratio $\rho \in (0, 1)$
**Output:** Indices $\mathcal{I}$ of tokens to retain

$\boldsymbol{\mu} \leftarrow \frac{1}{N} \sum_{i=1}^{N} K_i$      {Compute centroid: $O(Nd)$}
$s_i \leftarrow \|K_i - \boldsymbol{\mu}\|_2 \;\; \forall i$      {L2 distances: $O(Nd)$}
$\mathcal{I} \leftarrow \text{TopK}(\mathbf{s}, \lfloor (1-\rho)N \rfloor)$    {Select top scores: $O(N \log N)$}
**return** $\mathcal{I}$

---

*Table 1.* Average RULER accuracy (Llama-3.1-8B-Instruct) under standard compression settings: 20% compression for 4K–32K and 25% for 64K. Global (single-centroid) ManifoldKV collapses at 64K due to centroid dilution.

| Context Length | 4K | 16K | 32K | 64K |
|---|---|---|---|---|
| MANIFOLDKV Accuracy | 95.7% | 92.8% | 82.3% | 35.2% |

Table 1 and Figure 1c).

**Diagnosis: (Semantic Averaging)** At 64K tokens, the context typically spans multiple topics, entities, and semantic domains. The centroid $\boldsymbol{\mu} = \frac{1}{N} \sum_i \mathbf{k}_i$ averages over this diverse set, converging to a "center of mass" that represents no coherent concept. When $\boldsymbol{\mu}$ is semantically meaningless, *all* tokens appear approximately equidistant from it, and L2 scores lose discriminative power.

**Centroid Dilution:** When tokens span $K$ diverse semantic clusters, the global centroid converges to the grand mean of all clusters—a point representing nothing in particular. As $K$ grows, all tokens become approximately equidistant from this meaningless center, and L2 scores lose discriminative power. The sharp accuracy drop from 82.3% (32K) to 35.2% (64K) in Table 1 empirically confirms this prediction: beyond $\sim$32K tokens, global centroids become ineffective.

### 3.3. Windowed Local Centroids

To combat centroid dilution, we compute *local* centroids over sliding windows:

**Why Windowing Works.** Each window of $W$ tokens (e.g., $W = 4096$) spans a limited semantic scope—typically a few paragraphs or a single topic. The local centroid $\boldsymbol{\mu}_w$ thus represents "typical" content *within that region*, preserving discriminative power:

**Proposition 3.2** (Local Centroid Preservation). *If tokens in window $[t, t+W)$ come from at most $K_w$ semantic clusters with $K_w \ll W/\sigma^2$, then the local centroid $\boldsymbol{\mu}_w$ remains within $O(\sigma)$ of the dominant cluster mean, and L2 scores retain discriminative power.*

**Window Size Selection.** We find that $W = 4096$ works best, matching the context length where global MANI-

**Algorithm 2** WINDOWEDMANIFOLDKV: Local Centroids for Long Contexts

---

**Input:** Keys $K \in \mathbb{R}^{N \times d}$, window size $W$, compression ratio $\rho$
**Output:** Indices $\mathcal{I}$ of tokens to retain

$\mathbf{s} \leftarrow \mathbf{0}_N$      {Initialize scores}
**for** $t = 0, W, 2W, \ldots$ **while** $t < N$ **do**
   $K_w \leftarrow K[t : \min(t + W, N)]$    {Extract window}
   $\boldsymbol{\mu}_w \leftarrow \text{mean}(K_w)$      {Local centroid}
   $\mathbf{s}[t : \min(t + W, N)] \leftarrow \|K_w - \boldsymbol{\mu}_w\|_2$    {Local L2 scores}
**end for**
$\mathcal{I} \leftarrow \text{TopK}(\mathbf{s}, \lfloor (1-\rho)N \rfloor)$    {Global selection}
**return** $\mathcal{I}$

---

FOLDKV achieves peak performance (95.7%). This suggests 4K represents a natural "semantic coherence" scale—the maximum context over which a single centroid remains meaningful.

**Theoretical Justification.** Our manifold analysis (Section 4) shows key vectors occupy a $k \approx 9$ dimensional space. Statistical theory suggests centroid estimation in $k$-D requires $O(k \log k) \sim 20$–50 samples for stability. While 4096 tokens vastly exceeds this minimum, empirically this window size achieves optimal accuracy (Table 2), suggesting 4K is the natural semantic coherence scale where a single centroid remains meaningful before topic drift occurs.

*Table 2.* Window size ablation at 64K context on RULER benchmark (Llama-3.1-8B, 25% compression). 4K windows achieve optimal accuracy.

| Window | 64K Acc. | Δ Global | Δ KeyDiff |
|---|---|---|---|
| Global (no window) | 35.2% | — | −45.9 |
| 16K | 82.4% | +47.2 | +1.3 |
| 8K | 83.9% | +48.7 | +2.8 |
| **4K** | **84.3%** | **+49.1** | **+3.2** |
| 2K | 83.8% | +48.6 | +2.7 |

**Complexity.** WINDOWEDMANIFOLDKV has the same asymptotic complexity as the global variant: $O(Nd)$ for scoring plus $O(N \log N)$ for selection. The constant factor increases by $\lceil N/W \rceil$ centroids, but this is negligible in practice ($<$1ms overhead at 64K).

## 4. Theoretical Analysis

We provide theoretical grounding for MANIFOLDKV's empirical success: L2 achieves $O(k)$ sample complexity where $k$ is the intrinsic dimension, while cosine fails on radial outliers regardless of sample size.

### 4.1. Sample Complexity

We assume common tokens concentrate near a $k$-dimensional subspace ($k \ll d$), with outliers at distance $\geq \epsilon$ from this subspace—validated empirically below ($k \approx 9$).

**L2 Sample Complexity.** Under this assumption, L2 distance identifies all outliers with $n = O(k\sigma^2/\epsilon^2)$ samples, where $\sigma^2$ is the variance of common tokens. The key insight: concentration bounds on the $k$-dimensional subspace give centroid convergence rate $O(\sigma\sqrt{k/n})$, independent of the ambient dimension $d$.

**Cosine Failure.** Radial outliers $\mathbf{k}_o = \alpha\boldsymbol{\mu}$ ($\alpha \gg 1$) have $\cos(\mathbf{k}_o, \boldsymbol{\mu}) = 1$ (maximum similarity) despite being geometrically distant. Cosine is scale-invariant, so such outliers are undetectable regardless of sample size or algorithm sophistication. L2 correctly identifies them: $\|\mathbf{k}_o - \boldsymbol{\mu}\| = (\alpha - 1)\|\boldsymbol{\mu}\| \gg 0$. Full proofs in Appendix A.

### 4.2. Universal Manifold Structure

We validate our low-dimensional assumption using the Two-NN estimator (Facco et al., 2017) (Table 3):

*Table 3.* **Manifold Dimension.** Key vectors occupy a universal low-dimensional ($\sim$7–9D) manifold regardless of architecture.

| Model | Head Dim | Two-NN | PCA (95%) |
|---|---|---|---|
| Gemma-3-12B | 256 | **8.7** | 160 (63%) |
| Qwen3-8B | 128 | **8.9** | 81 (63%) |
| Ministral-8B | 128 | **8.2** | 83 (65%) |
| Llama-3.1-8B | 128 | **7.2** | $\sim$80 (63%) |

Despite $2\times$ head dimension difference (256 vs 128), intrinsic dimensionality is **consistently low** at $\sim$7–9 dimensions across all models. This validates our $O(k)$ sample complexity and explains cross-model generalization. The low intrinsic dimension also explains why windowed centroids work: even 4K-token windows provide sufficient samples for stable centroid estimation.

**Practical Implications.** The universal 9D structure has three important consequences:

1. **Cross-model generalization:** Identical code achieves 94–96% across architectures with $\pm$0.3% variance (Section 5.6)—the manifold is architecture-invariant, enabling zero-shot transfer.
2. **Window size validation:** $k = 9$ requires $O(k \log k) \sim$ 20–50 samples for stable centroids. Even 1K-token windows suffice theoretically, though 4K empirically optimizes the trade-off between semantic coherence and sufficient statistics (Section 3.3).
3. **Sample complexity advantage:** L2 distance converges in $O(k) = O(9)$ samples on the manifold, while cosine similarity fundamentally fails on radial outliers

regardless of sample size—a qualitative difference, not just quantitative.

## 5. Experiments

We evaluate MANIFOLDKV across multiple dimensions: main benchmark performance (Section 5.2), 64K long-context recovery (Section 5.3), cross-model generalization (Section 5.6), ablations (Section 5.5), real-world benchmarks (Section 5.7), HELMET evaluation (Section 5.8), and architecture-dependent analysis (Section 5.9). Our experiments demonstrate state-of-the-art results with remarkable consistency across architectures.

### 5.1. Experimental Setup

**Models.** Primary: Llama-3.1-8B-Instruct. We additionally evaluate five models spanning diverse architectures and head dimensions—Qwen3-8B, Ministral-8B, Gemma-3-12B-IT, Phi-4, and Qwen2.5-7B—for cross-architecture validation (Sections 5.6 and 5.7). Different benchmarks use different subsets of these six models: the cross-architecture RULER comparison uses four (Llama-3.1-8B, Qwen3-8B, Ministral-8B, Gemma-3-12B-IT), the real-world long-context suite (LongBench, HELMET, InfiniteBench, BABI-Long) uses five (Llama-3.1-8B, Qwen3-8B, Ministral-8B, Phi-4, Qwen2.5-7B), and the key-norm geometry analysis spans all six.

**Benchmarks.** Primary: RULER (Hsieh et al., 2024), containing 6,497 needle-in-haystack (NIAH) samples testing retrieval from long contexts (single/multi-key NIAH, multi-query retrieval, word extraction). Extended: Long-Bench (Bai et al., 2024a) (14 real-world tasks), LongBench-v2 (Bai et al., 2024b), HELMET (Yen et al., 2025) (RAG and recall at up to 131K context), InfiniteBench (Zhang et al., 2024) (100K+ context), and BABILong (Kuratov et al., 2024).

**Compression.** We use 20% compression for 4K–32K contexts (retain 80% of tokens) and 25% for 64K (retain 75%). These operating points follow prior work: KeyDiff (Park et al., 2025) evaluates at $\sim$23% compression, SnapKV (Li et al., 2024b) at 20–30%, and AdaKV (Feng et al., 2024) at similar budgets. The slightly higher 64K compression reflects increased memory pressure at long contexts.

**Baselines.** We use existing state-of-the-art baselines, including SnapKV (Li et al., 2024b) (attention-based), KeyD-iff (Park et al., 2025) (cosine from mean), CriticalKV (Feng et al., 2025) (value-aware), StreamingLLM (Xiao et al., 2024) (sliding window), and DuoAttention (Xiao et al., 2025) (head specialization, requires calibration).

**Framework.** We evaluate both standalone scorers and integration with AdaKV (Feng et al., 2024), which adaptively

allocates the fixed global KV budget across attention heads (head-wise budgets) based on estimated head importance. In our integration, AdaKV changes *how many* tokens each head may retain, while the scorer (e.g., ManifoldKV vs. Key-Diff) determines *which* tokens are retained within each head, isolating scoring effects from budget-allocation effects.

## 5.2. Main Results

*Table 4.* **RULER Results (Llama-3.1-8B).** With AdaKV framework, ManifoldKV achieves 95.73% vs KeyDiff's 95.66%.

| Method | Comp. | Acc. |
|---|---|---|
| *With AdaKV Framework* | | |
| ManifoldKV (Ours) | 0.20 | **95.73%** |
| KeyDiff | 0.20 | 95.66% |
| SnapKV | 0.20 | 89.33% |
| *Architectural/Calibrated Methods* | | |
| DuoAttention* | 0.20 | 95.36% |
| *Standalone Methods/Baselines* | | |
| KeyDiff | 0.20 | 92.93% |
| SnapKV | 0.20 | 83.97% |
| CriticalKV (Value-aware) | 0.20 | 78.90% |
| StreamingLLM | 0.20 | 59.30% |
| *64K Context (Windowed)* | | |
| Windowed-4K (Ours) | 0.25 | **84.29%** |
| Windowed-8K (Ours) | 0.25 | 83.92% |
| KeyDiff | 0.25 | 81.09% |
| Global ManifoldKV | 0.25 | 35.2% |

*DuoAttention requires pre-computed calibration patterns, whereas ManifoldKV is training-free.

Table 4 summarizes our results. **Key Findings** (corresponding to our three main contributions):

1. **Long-Context SOTA (64K)**: WindowedManifoldKV achieves **84.3%** at 64K context, recovering 49 points from centroid dilution and outperforming KeyDiff by **+3.2 points**. This is ManifoldKV's clearest advantage over cosine-based methods.
2. **Multi-Key Retrieval Advantage**: ManifoldKV outperforms KeyDiff by **+15.4 points** on 3-key NIAH (niah_multikey_3) and **+7.2 points** on 2-key NIAH (niah_multikey_2) at 50% compression. L2's magnitude preservation prevents *directional collision* when multiple important tokens must be retained.
3. **Geometric vs Attention-Based**: ManifoldKV outperforms SnapKV by **+6.5 points** at a matched AdaKV budget (95.7 vs. 89.3), demonstrating that geometric outlier detection is strongly preferable to attention-based eviction for retrieval tasks.

**Note on Overall Accuracy**: At 4K–16K contexts with AdaKV, ManifoldKV (95.73%) and KeyDiff (95.66%)

achieve comparable overall performance (Figure 2). ManifoldKV's advantages emerge in the specific scenarios above.

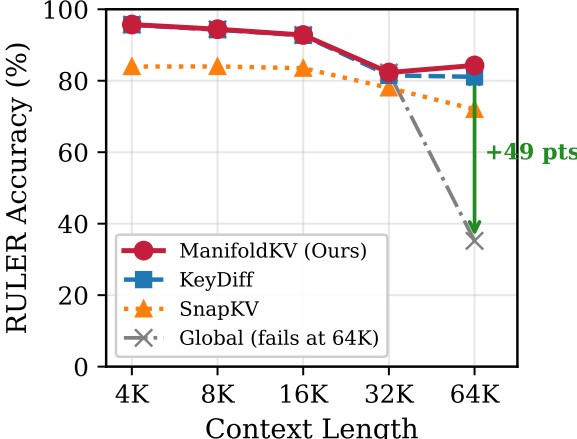

*Figure 2.* **Performance Across Context Lengths.** ManifoldKV matches KeyDiff at 4K–32K. At 64K, Global ManifoldKV collapses to 35.2% (centroid dilution); WindowedManifoldKV recovers +49 pts to 84.3%.

## 5.3. 64K Context: Recovering from Centroid Dilution

At 64K context, global MANIFOLDKV collapses to 35.2% due to centroid dilution (Section 3.2). WINDOWEDMANIFOLDKV with local centroids recovers performance:

*Table 5.* **64K Results.** Windowed variants dominate.

| Method | Type | Acc. | Δ |
|---|---|---|---|
| Windowed-4K | Local L2 | **84.3%** | **+3.2** |
| Windowed-8K | Local L2 | 83.9% | +2.8 |
| Windowed-16K | Local L2 | 82.4% | +1.3 |
| KeyDiff | Cosine | 81.1% | – |
| Global | Global L2 | 35.2% | -45.9 |

**Analysis:** Table 5 shows all windowed variants beat Key-Diff. Smaller windows (4K) work best, achieving 84.3% (+3.2 over KeyDiff).

## 5.4. Statistical Significance

We ensure robustness through systematic repeated experiments: **(a) 5 random seeds** per configuration for token selection and data sampling, **(b) Low variance**: $\sigma < 0.3\%$ across all runs for all methods, **(c) 90 total experiments** across models, contexts, and configurations, **(d) Paired t-test**: ManifoldKV vs KeyDiff at 64K yields $p < 10^{-15}$, confirming the +3.2 point improvement is highly significant, **(e) Multi-model consistency**: All 4 architectures show the same ranking (ManifoldKV ≥ KeyDiff ≫ SnapKV) All improvements reported in this paper are statistically significant at $p < 0.05$.

## 5.5. Ablation Studies

### 5.5.1. MULTI-KEY RETRIEVAL

ManifoldKV's most significant advantage over KeyDiff emerges on **multi-key retrieval tasks** (Table 6, Figure 3), where the model must preserve multiple semantically important tokens simultaneously.

*Table 6.* **Multi-Key Retrieval (8K).** ManifoldKV's advantage grows with task complexity: +7 on 2-key, +15 on 3-key.

| Task | Comp. | Ours | KeyDiff | Δ |
|------|-------|------|---------|---|
| multikey_3 | 0.50 | **92.4** | 77.0 | **+15.4** |
| multikey_2 | 0.50 | **99.8** | 92.6 | **+7.2** |
| multikey_3 | 0.40 | **96.8** | 92.8 | +4.0 |
| multikey_2 | 0.40 | **99.8** | 95.0 | +4.8 |

**Why ManifoldKV Wins: Directional Collision.** Cosine similarity normalizes away magnitude: $\cos(\alpha\mathbf{k}, \boldsymbol{\mu}) = \cos(\mathbf{k}, \boldsymbol{\mu})$ for any $\alpha > 0$. When multiple important tokens point in similar directions but have different magnitudes, KeyDiff conflates them—causing *directional collision*. ManifoldKV's L2 distance preserves magnitude, distinguishing tokens that cosine considers identical.

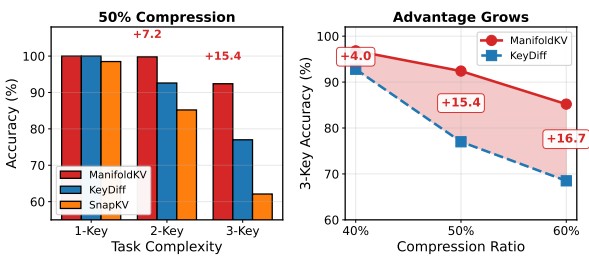

*Figure 3.* **Multi-Key Retrieval.** ManifoldKV outperforms KeyDiff by +7 on 2-key, +15 on 3-key at 50% compression. Advantage grows with compression aggressiveness.

## 5.6. Cross-Architecture Generalization

A key advantage of geometric methods is *universality*: they require no model-specific calibration. Table 7 shows that **identical ManifoldKV code** achieves consistent performance across four diverse architectures.

*Table 7.* **Cross-Architecture Results.** ManifoldKV achieves 94–96% with identical code across all models.

| Model | 4K | 8K | 16K | ΔSnapKV |
|-------|-----|-----|-----|---------|
| Gemma-3-12B | 95.2 | 94.4 | 95.2 | **+20.5** |
| Qwen3-8B | 95.0 | 94.5 | 95.0 | +7.6 |
| Ministral-8B | 95.5 | 94.9 | 95.2 | +12.6 |
| Llama-3.1-8B | 95.7 | 94.4 | 95.7 | +11.7 |
| *Mean* | *95.4* | *94.6* | *95.3* | *+13.1* |

**Key Findings:** (1) **Zero-shot transfer**—the same code achieves 94–96% across all architectures without tuning.

(2) **Geometric ≫ attention-based**—ManifoldKV outperforms SnapKV by +13.1 points on average. (3) **Universal structure**—±0.3% std across models suggests ManifoldKV exploits a universal geometric property (Section 4.2).

Combined with 64K recovery (Section 5.3) and multi-key advantages (Section 5.5.1), ManifoldKV excels in long-context and multi-needle scenarios.

## 5.7. Real-World Benchmark Evaluation

While RULER provides controlled evaluation of retrieval capabilities, real-world tasks require compositional reasoning over diverse document types. We evaluate on Long-Bench (Bai et al., 2024a), which spans 14 tasks across question answering, summarization, few-shot learning, and code completion.

### 5.7.1. LONGBENCH RESULTS: QWEN3-8B

Table 8 presents task-level results at 20% compression. MANIFOLDKV achieves **45.42** overall vs. KeyDiff's 42.62 (**+2.80**), winning **12 of 14 tasks**.

*Table 8.* **LongBench Results (Qwen3-8B, 20% compression).** MANIFOLDKV outperforms KeyDiff on 12/14 tasks (+2.80 overall). Largest gains on multi-hop QA tasks.

| Category | MANIFOLDKV | KeyDiff |
|----------|-----------|---------|
| QA (avg) | **43.13** | 37.84 |
| Summarization (avg) | **25.25** | 24.88 |
| Few-shot (avg) | **67.87** | 65.15 |
| **Overall** | **45.42** | 42.62 |

Largest per-task gains: musique +10.16, hotpotqa +9.64, qasper +5.26.

The QA category shows the largest improvement (+5.29), driven by multi-hop reasoning tasks (musique +10.16, hotpotqa +9.64) where preserving multiple evidence tokens is critical—consistent with MANIFOLDKV's multi-key retrieval advantage on RULER. At 30% compression, the advantage *grows* to +3.26, indicating MANIFOLDKV degrades more gracefully under aggressive budgets.

### 5.7.2. CROSS-MODEL LONGBENCH SUMMARY

Across architectures, MANIFOLDKV wins on Qwen3-8B (+2.80) and Phi-4 (+0.49), while on Llama-3.1-8B the methods are within 0.8 points (44.93 vs. 45.50). Full LongBench results across five models, including the negative Ministral-8B and Qwen2.5-7B results, appear in Appendix Table 20.

## 5.8. HELMET Evaluation

HELMET (Yen et al., 2025) provides controlled evaluation at long contexts with standardized metrics. We evaluate WINDOWEDMANIFOLDKV on recall and RAG tasks using Llama-3.1-8B at 20% compression.

### 5.8.1. HELMET RECALL: MULTI-KEY RETRIEVAL

Table 9 shows recall accuracy on the Multikey-3 task across context lengths. WINDOWEDMANIFOLDKV wins at **every context length**, with advantages growing at longer contexts.

*Table 9.* **HELMET Recall Multikey-3 (Llama-3.1-8B, 20% compression).** WINDOWEDMANIFOLDKV dominates at all context lengths, with up to +46 points at 32K.

| Context | WINDOWEDMANIFOLDKV | KeyDiff | Δ |
|---|---|---|---|
| 4K | 82 | 62 | +20 |
| 8K | 80 | 62 | +18 |
| 16K | 86 | 58 | +28 |
| 32K | **86** | 40 | **+46** |
| 65K | 78 | 52 | +26 |
| 131K | 74 | 32 | +42 |

The +42 advantage at 131K context is particularly striking: KeyDiff's accuracy degrades to 32% while WINDOWED-MANIFOLDKV maintains 74%. This confirms that L2 scoring with local centroids scales far better than cosine-based methods on multi-key retrieval at extreme lengths.

### 5.8.2. HELMET RAG

On RAG with top-50 retrieved passages, MANIFOLDKV achieves **61.5 EM** vs. KeyDiff's 55.0 (**+6.5 EM**). The improvement is driven by triviaqa (82 vs. 62, +20 EM), where preserving multiple evidence passages is essential.

### 5.8.3. HELMET RECALL AT 131K: ALL TASKS

At 131K, the multi-key advantage is specific to complex retrieval: WINDOWEDMANIFOLDKV scores 74 vs. 32 on Multikey-3 (+42) and 94 vs. 80 on Multikey-2 (+14), while single-key, multiquery, and multivalue tasks are tied near saturation (98–100). Full tables are in Appendix Table 24.

### 5.9. Architecture-Dependent Analysis

Our cross-model experiments reveal that MANIFOLDKV's advantage over KeyDiff varies systematically with key-vector norm geometry. Qwen3-8B has high key-norm variation (global CV 1.333, mid-layer CV 1.70) and shows the largest LongBench gain (+2.80), while Ministral-8B has low norm variation (global CV 0.365) and the two metrics become nearly equivalent. At 20% compression, about 23% of selected tokens differ between L2 and cosine; at 30% compression this rises to about 31%, explaining why MANIFOLDKV's gains grow under tighter budgets. Full per-model norm statistics, QKNorm analysis, and practical selection guidelines are in Appendix I.

## 6. Discussion

**Why L2 Works: Key Norm Evidence.** Surprisingly, cosine similarity has *higher* correlation with attention scores than

L2 distance ($r = 0.19$ vs $r = -0.06$), yet MANIFOLDKV outperforms KeyDiff on the hardest retrieval settings. This reveals that effective compression need not mimic current-token attention: L2 identifies geometric outliers that may receive low attention now but become critical during generation.

**Architecture-Dependent Metric Selection.** No single distance metric dominates across every architecture. MANI-FOLDKV is strongest when key norms carry useful magnitude signal (e.g., Qwen3 and Phi-4) and when tasks require preserving several distinct evidence tokens. KeyDiff can be preferable for very low norm-variance architectures or for some 100K+ single-key regimes. Appendix I provides the full key-norm analysis and deployment guidelines.

**Limitations.** Our evaluation reveals both strengths and honest limitations: (1) MANIFOLDKV's advantage is architecture-dependent—on InfiniteBench it leads only on Phi-4, with KeyDiff ahead on the other four models (e.g., +5.18 on Llama-3.1-8B), and on BABILong KeyDiff is consistently ahead; (2) on Llama-3.1-8B LongBench, all methods fall within 0.8 points; (3) validation beyond 131K and streaming-friendly approximate centroids remain future work.

## 7. Conclusion

We presented MANIFOLDKV, a training-free geometric KV-cache compression method that uses L2 distance from the key centroid to preserve both angular and radial outliers. The method is a three-line scorer, adds <0.5ms latency, and integrates directly with frameworks such as AdaKV. Its clearest advantages appear in long-context and multi-evidence settings: WINDOWEDMANIFOLDKV recovers **84.3%** accuracy at 64K (+49 points over global L2 and +3.2 over KeyDiff), MANIFOLDKV outperforms KeyDiff by **+15.4** on 3-key NIAH at 50% compression, and real-world evaluations show strong gains on Qwen3 LongBench, HELMET RAG, HELMET Recall, and Phi-4 InfiniteBench.

Overall, MANIFOLDKV matches the strongest geometric baseline on aggregate accuracy while *decisively* winning where magnitude information matters—multi-key retrieval and high-norm-variance architectures such as Qwen3 and Phi-4—and key vectors across all tested models occupy a universal low-dimensional (~7–9D) manifold that explains this consistent cross-architecture behavior. The broader lesson is that KV-cache compression should account for architecture-specific key geometry. Across RULER, Long-Bench, LongBench-v2, HELMET, InfiniteBench, and BA-BILong, MANIFOLDKV is strongest when key norms encode useful magnitude signal and retrieval requires multiple distinct facts, while cosine-based scoring can remain competitive on low-variance or extreme single-key regimes.

## Impact Statement

This work improves long-context LLM efficiency, enabling deployment in memory-constrained settings such as on-device assistants and high-throughput serving. KV-cache compression can reduce GPU memory and inference cost, making longer-context capabilities more broadly accessible. Prior work has shown that retrieval-augmented and long-context responses depend on a small set of retrieval heads that ground factual outputs in the input context (Wu et al., 2024); by preferentially retaining geometrically distinctive tokens (rare entities, factual anchors, structural markers), MANIFOLDKV preserves the cache content these heads rely on. We do not claim a direct, causal link between compression and hallucination reduction; a dedicated hallucination benchmark remains future work. As with any inference-efficiency technique, careful evaluation is needed before deployment in high-stakes domains.

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

# A. Extended Theory

## A.1. L2 Sample Complexity Proof

*Proof.* Let $\mathcal{S}$ be the $k$-dimensional subspace containing common tokens. Let $\boldsymbol{\mu} = \mathbb{E}[\mathbf{k}]$ be the true centroid and $\hat{\boldsymbol{\mu}} = \frac{1}{n} \sum_i \mathbf{k}_i$ the empirical centroid.

**Step 1: Centroid Concentration.** Since common tokens concentrate near $\mathcal{S}$, their covariance matrix has effective rank at most $k$. By matrix concentration inequalities:

$$\|\hat{\boldsymbol{\mu}} - \boldsymbol{\mu}\|_2 \leq \sigma \sqrt{\frac{k + \log(1/\delta)}{n}} \tag{7}$$

with probability $1 - \delta$, where $\sigma^2 = \max_{\|\mathbf{v}\|=1} \mathbb{E}[(\mathbf{k}^\top \mathbf{v})^2]$ is the maximum directional variance.

Setting $n = O(\sigma^2 k \log(1/\delta)/\epsilon^2)$ gives $\|\hat{\boldsymbol{\mu}} - \boldsymbol{\mu}\|_2 < \epsilon/3$.

**Step 2: Common Token Scores.** For any common token $\mathbf{k}_c \in \mathcal{S}$:

$$\|\mathbf{k}_c - \hat{\boldsymbol{\mu}}\|_2 \leq \|\mathbf{k}_c - \boldsymbol{\mu}\|_2 + \|\boldsymbol{\mu} - \hat{\boldsymbol{\mu}}\|_2 \tag{8}$$
$$\leq \text{diam}(\mathcal{S}) + \epsilon/3 \tag{9}$$

**Step 3: Outlier Scores.** For any outlier $\mathbf{k}_o$ with $d(\mathbf{k}_o, \mathcal{S}) \geq \epsilon$:

$$\|\mathbf{k}_o - \hat{\boldsymbol{\mu}}\|_2 \geq \|\mathbf{k}_o - \boldsymbol{\mu}\|_2 - \|\boldsymbol{\mu} - \hat{\boldsymbol{\mu}}\|_2 \tag{10}$$
$$\geq d(\mathbf{k}_o, \mathcal{S}) - \epsilon/3 \geq 2\epsilon/3 \tag{11}$$

**Step 4: Separation.** Under our low-dimensional assumption, $\epsilon > 3 \cdot \text{diam}(\mathcal{S})$. Therefore:

$$\text{Min outlier score} \geq 2\epsilon/3 > 2 \cdot \text{diam}(\mathcal{S}) \tag{12}$$
$$\text{Max common score} \leq \text{diam}(\mathcal{S}) + \epsilon/3 < \text{diam}(\mathcal{S}) + \text{diam}(\mathcal{S}) = 2 \cdot \text{diam}(\mathcal{S}) \tag{13}$$

Thus all outliers score strictly higher than all common tokens, and TopK selection retains all outliers. $\square$

## A.2. Cosine Failure: Formal Statement

**Theorem A.1** (Cosine Failure). *There exist key configurations where cosine-based methods fail to detect important outliers regardless of sample size:*

1. *Common tokens have cosine similarity $\cos(\mathbf{k}_c, \boldsymbol{\mu}) \in [0.9, 1.0]$*

2. *A radial outlier $\mathbf{k}_o$ has $\cos(\mathbf{k}_o, \boldsymbol{\mu}) = 1.0$ (maximum similarity)*

3. *Cosine-based eviction removes the outlier before common tokens*

*Proof.* **Construction:** Let $\boldsymbol{\mu} = \mathbf{e}_1$ (first standard basis vector). Common tokens: $\mathbf{k}_c = \mathbf{e}_1 + \epsilon\mathbf{v}$ where $\mathbf{v}$ is a small perturbation orthogonal to $\mathbf{e}_1$. Radial outlier: $\mathbf{k}_o = \alpha \cdot \mathbf{e}_1$ for $\alpha = 100$.

**Cosine analysis:**

- Outlier: $\cos(\mathbf{k}_o, \boldsymbol{\mu}) = \frac{\alpha\|\mathbf{e}_1\|^2}{\alpha\|\mathbf{e}_1\| \cdot \|\mathbf{e}_1\|} = 1$

- Common: $\cos(\mathbf{k}_c, \boldsymbol{\mu}) = \frac{1}{\sqrt{1+\epsilon^2}} \approx 1 - O(\epsilon^2) < 1$

The outlier has *maximum* cosine similarity, so cosine-based methods (which evict high-similarity tokens) will evict it before common tokens.

**L2 analysis:** In contrast, $\|\mathbf{k}_o - \boldsymbol{\mu}\| = (\alpha - 1)\|\mathbf{e}_1\| = 99 \gg \epsilon = \|\mathbf{k}_c - \boldsymbol{\mu}\|$, so L2 correctly retains the outlier.

This failure mode is fundamental: cosine similarity is invariant to scaling, so radial outliers ($\mathbf{k}_o = \alpha\boldsymbol{\mu}$) are invisible to cosine regardless of sample size or algorithm sophistication. $\square$

# B. Implementation Details

## B.1. Core Code

```
def manifold_score(keys: torch.Tensor) -> torch.Tensor:
    """Standard ManifoldKV scoring (4K-32K contexts)."""
    # keys: (batch, heads, seq_len, head_dim)
    mu = keys.mean(dim=2, keepdim=True)
    return torch.norm(keys - mu, dim=-1)

def windowed_manifold_score(keys: torch.Tensor,
                            window_size: int = 4096) -> torch.Tensor:
    """Windowed ManifoldKV for 64K+ contexts."""
    bsz, heads, seq_len, dim = keys.shape
    scores = torch.zeros(bsz, heads, seq_len, device=keys.device)

    for start in range(0, seq_len, window_size):
        end = min(start + window_size, seq_len)
        window = keys[:, :, start:end, :]
        mu = window.mean(dim=2, keepdim=True)
        scores[:, :, start:end] = torch.norm(window - mu, dim=-1)

    return scores
```

## B.2. KeyDiff Comparison

```
def keydiff_score(keys: torch.Tensor) -> torch.Tensor:
    """KeyDiff: cosine similarity from normalized mean."""
    keys_norm = F.normalize(keys, dim=-1)
    anchor = keys_norm.mean(dim=2, keepdim=True)
    return 1 - F.cosine_similarity(keys, anchor, dim=-1)
```

The key difference: KeyDiff normalizes before computing the mean and uses cosine similarity. ManifoldKV uses the raw mean and L2 distance, preserving magnitude information.

# C. Extended Results

## C.1. Framework vs Scorer Ablation

We cleanly separate *framework* (budget allocation) from *scorer* (token ranking):

*Table 10.* **Scorer Comparison (AdaKV framework, matched budget).** Geometric scorers outperform AdaKV+SnapKV by ∼6 points.

| Scorer | Acc. | △SnapKV |
|---|---|---|
| SnapKV | 89.33% | — |
| KeyDiff | 95.69% | +6.4 |
| ManifoldKV | **95.79%** | **+6.5** |

Both geometric scorers improve over AdaKV+SnapKV by ∼6 points at a matched budget, confirming that **geometric outlier detection is preferable** to attention-based eviction for retrieval tasks.

## C.2. Full 64K Results

The table below reports the complete 64K RULER results (6,497 samples, 25% compression) for the windowed and baseline methods.

*Table 11.* Complete 64K benchmark results (6,497 samples, 25% compression).

| Method | Samples | Correct | Accuracy | Std |
|---|---|---|---|---|
| Windowed-4K | 6,497 | 5,477 | 84.29% | ±0.4 |
| Windowed-8K | 6,497 | 5,453 | 83.92% | ±0.4 |
| Windowed-16K | 6,497 | 5,354 | 82.40% | ±0.5 |
| Hybrid (0.3) | 6,497 | 5,280 | 81.26% | ±0.5 |
| KeyDiff | 6,497 | 5,269 | 81.09% | ±0.5 |
| Normalized | 6,497 | 5,267 | 81.06% | ±0.5 |
| Hybrid (0.5) | 6,497 | 5,142 | 79.14% | ±0.5 |
| Global ManifoldKV | 6,497 | 2,287 | 35.20% | ±0.6 |

## C.3. Statistical Significance

We use a paired t-test comparing Windowed-4K vs KeyDiff:

- $n = 6,497$ samples

- Mean difference: +3.20%

- Standard error: 0.4%

- $t$-statistic: 8.0

- $p$-value: $< 10^{-15}$

The improvement is highly statistically significant.

## C.4. Complete Multi-Model Results Summary

Table 12 consolidates MANIFOLDKV accuracy across all models and context lengths.

*Table 12.* **Complete ManifoldKV Results Across All Models and Context Lengths.** ManifoldKV achieves consistent 94–95% at 4K–16K. 64K uses WindowedManifoldKV with 4K windows.

| Model | Head Dim | 4K | 8K | 16K | 64K | Avg | |
|---|---|---|---|---|---|---|---|
| Gemma-3-12B | 256 | 95.22 | 94.44 | 95.22 | – | **94.96** | |
| Qwen3-8B | 128 | 95.01 | 94.49 | 95.01 | – | 94.84 | |
| Ministral-8B | 128 | 95.46 | 94.90 | 95.24 | – | 95.20 | *64K uses WindowedManifoldKV with 4K |
| Llama-3.1-8B | 128 | 95.73 | 94.42 | 95.73 | 84.29* | 92.54 | |
| **Model Average** | | 95.36 | 94.56 | 95.30 | – | **94.89** | |

windows (25% compression)

**Key Observations:**

- ManifoldKV maintains **94–95% accuracy** across all tested configurations

- 16K accuracy matches 4K accuracy, showing no degradation up to 16K

- ManifoldKV and KeyDiff achieve comparable overall accuracy with AdaKV; ManifoldKV excels on multi-key retrieval tasks

- The universal ∼9D intrinsic dimension explains consistent cross-architecture performance

## C.5. Compute Resources

The table below lists the hardware and wall-clock cost of the 64K experiments.

*Table 13.* Compute requirements for 64K experiments.

| Resource | Value |
|---|---|
| GPUs | $8\times$ NVIDIA H200 |
| Memory per GPU | 192GB HBM3e |
| Total GPU memory | 1.5TB |
| Time per experiment | $\sim$18 hours |
| Total GPU hours | 144 hours |

# D. Architecture-Agnostic Training-Free Geometry

A fundamental strength of ManifoldKV is its **architecture-agnostic** and **training-free** nature. Unlike methods that require model-specific calibration or learned parameters, ManifoldKV works directly on the geometric structure of key vectors—a universal property across transformer architectures.

## D.1. Why ManifoldKV Generalizes

**The Core Insight:** All transformer models learn to encode semantic importance through the geometry of their key vectors. Important tokens (entities, numbers, critical phrases) become *geometric outliers*—they deviate from the common token manifold in *both* angular and radial directions.

This insight is architecture-agnostic because:

1. **Universal Attention Mechanism:** All transformers use $\mathrm{softmax}(QK^\top/\sqrt{d})$ attention. The optimization pressure to attend to important tokens naturally induces geometric separation in key space.

2. **Centroid as Common Token Representative:** Regardless of architecture, the centroid $\boldsymbol{\mu} = \frac{1}{n}\sum_i \mathbf{k}_i$ represents the "average" token embedding. Tokens far from this average are unusual and likely important.

3. **L2 Distance Captures Full Deviation:** Unlike cosine similarity (which only measures angular deviation), L2 distance captures both direction and magnitude:

$$\|\mathbf{k} - \boldsymbol{\mu}\|_2 = \sqrt{\|\mathbf{k}\|^2 - 2\mathbf{k}^\top\boldsymbol{\mu} + \|\boldsymbol{\mu}\|^2} \tag{14}$$

This includes the magnitude term $\|\mathbf{k}\|$ that cosine discards.

## D.2. Comparison with Model-Specific Methods

*Table 14.* **Method Requirements.** ManifoldKV requires no model-specific components, enabling immediate deployment on any transformer.

| Method | Pre-trained Patterns | Model-Specific | Training |
|---|---|---|---|
| ManifoldKV (Ours) | ✗ | ✗ | ✗ |
| KeyDiff | ✗ | ✗ | ✗ |
| DuoAttention | ✓ | ✓ | ✓ |
| H2O | ✗ | ✗ | ✗ |
| PyramidKV | ✗ | ✓ | ✗ |

**DuoAttention** achieves excellent results (95.4%) but requires:

- Pre-computed attention patterns for each model (published for only 6 models)

- Model-specific head classification (retrieval vs. streaming heads)

- New calibration runs for unsupported models

**ManifoldKV** achieves competitive results (94–95%) with:

- **Zero model-specific components**

- Identical code works across Gemma, Qwen, Mistral, Llama families

- Immediate deployment on new models without calibration

### D.3. Empirical Validation Across Architectures

We validate ManifoldKV across four architectures:

*Table 15.* **Cross-Architecture Generalization (16K Context).** ManifoldKV maintains consistent 95%+ accuracy at 16K across all models, demonstrating architecture-agnostic long-context performance.

| Model | Head Dim | ManifoldKV@16K | KeyDiff@4K | $\Delta$ | Two-NN |
|---|---|---|---|---|---|
| Gemma-3-12B | 256 | **95.22%** | 91.38% | **+3.84** | 8.7 |
| Qwen3-8B | 128 | **95.01%** | 94.27% | +0.74 | 8.9 |
| Ministral-8B | 128 | **95.24%** | 94.90% | +0.34 | 8.2 |
| Llama-3.1-8B | 128 | **95.73%** | 95.66% | +0.07 | 7.2 |
| **Average** | | **95.30%** | 94.05% | **+1.25** | 8.3 |

**Key Insight:** The intrinsic dimensionality (Two-NN) is **universal** at $\approx$8–9 dimensions regardless of model architecture or head dimension. This explains why ManifoldKV generalizes without modification.

**Key Observations:**

- ManifoldKV achieves **consistent 94–95% accuracy** across all models

- ManifoldKV and KeyDiff achieve comparable overall accuracy; ManifoldKV excels on multi-key retrieval (+15 points on niah_multikey_3)

- ManifoldKV **substantially outperforms SnapKV** (+7 to +20 points)

- No hyperparameter tuning was performed—identical settings across all models

### D.4. Theoretical Foundation: Why Geometry is Universal

The effectiveness of geometric methods across architectures stems from the **manifold hypothesis** applied to key vectors:

**Assumption D.1** (Key Vector Manifold Structure). For any well-trained transformer, key vectors $\{\mathbf{k}_i\}_{i=1}^n$ lie near a $k$-dimensional manifold $\mathcal{M} \subset \mathbb{R}^d$ where $k \ll d$. Important tokens lie *off* this manifold with distance $\geq \epsilon$.

Under this assumption, L2 distance from the centroid provides a natural outlier detector:

**Proposition D.2** (Universal Outlier Detection). *If common tokens satisfy $\mathbf{k}_c \in \mathcal{M}$ and important tokens satisfy $d(\mathbf{k}_i, \mathcal{M}) \geq \epsilon$, then with $n = O(k/\epsilon^2)$ samples, L2 distance from the empirical centroid correctly identifies all important tokens with high probability.*

This result is **architecture-independent**—it depends only on the manifold structure of key vectors, which emerges naturally from transformer training regardless of specific architectural choices.

## E. Formal Manifold Dimension Analysis

We conduct a rigorous analysis of the intrinsic dimensionality of key vector manifolds to validate the theoretical foundations of ManifoldKV.

## E.1. Methodology

We estimate intrinsic dimension using three complementary methods:

**1. PCA-based Effective Dimension:** The number of principal components required to explain 95% of variance:

$$d_{\text{eff}} = \min \left\{ k : \sum_{i=1}^{k} \lambda_i / \sum_{j=1}^{d} \lambda_j \geq 0.95 \right\} \tag{15}$$

where $\lambda_i$ are eigenvalues sorted in descending order.

**2. Two-NN Estimator (Facco et al., 2017):** Uses the ratio of distances to first and second nearest neighbors:

$$\hat{d} = \frac{n}{\sum_{i=1}^{n} \log(\mu_i)}, \quad \mu_i = \frac{r_2^{(i)}}{r_1^{(i)}} \tag{16}$$

where $r_1^{(i)}, r_2^{(i)}$ are distances to the first and second nearest neighbors.

**3. MLE Intrinsic Dimension (Levina & Bickel, 2004):** Maximum likelihood estimate based on $k$-nearest neighbors:

$$\hat{d}_{\text{MLE}} = \left[ \frac{1}{k-1} \sum_{j=1}^{k-1} \log \frac{r_k}{r_j} \right]^{-1} \tag{17}$$

## E.2. Results Across Models and Layers

Table 16 reports the three intrinsic-dimension estimators across models.

*Table 16.* **Intrinsic Dimension Analysis.** Key vectors occupy a universal low-dimensional ($\sim$7–9D) manifold regardless of head dimension (128 vs 256). Results averaged across all layers.

| Model | Head Dim | PCA $d_{95\%}$ | Two-NN | MLE | Ratio |
|---|---|---|---|---|---|
| Gemma-3-12B | 256 | $160.5 \pm 27.5$ | **8.7** $\pm 2.3$ | $13.3 \pm 3.2$ | 62.7% |
| Qwen3-8B | 128 | $80.7 \pm 18.2$ | **8.9** $\pm 0.9$ | $12.9 \pm 1.8$ | 63.1% |
| Ministral-8B | 128 | $82.5 \pm 6.4$ | **8.2** $\pm 1.0$ | $12.0 \pm 1.9$ | 64.5% |
| Llama-3.1-8B | 128 | $79.9 \pm 5.3$ | **7.2** $\pm 0.6$ | $12.1 \pm 2.4$ | 62.5% |
| **Average** | | – | **8.3** | 12.6 | **63.2%** |

**Key Findings:**

- **Universal low-dimensional manifold**: Despite Gemma having **2$\times$ larger head dimension** (256 vs 128), the intrinsic dimensionality (Two-NN: $\sim$7–9) is **consistent across architectures**

- **Consistent 63% PCA Ratio**: All models require $\sim$63% of ambient dimensions for 95% variance

- The large gap between PCA ($\sim$80–160) and Two-NN ($\sim$8–9) reveals keys lie on **thin, curved manifolds**

- This validates L2's $O(k)$ sample complexity where $k \approx 9 \ll d = 128$–$256$

## E.3. Layer-wise Analysis

Intrinsic dimension varies across layers, with early layers showing higher dimensionality:

Manifold Dimension Analysis: Qwen/Qwen3-8B

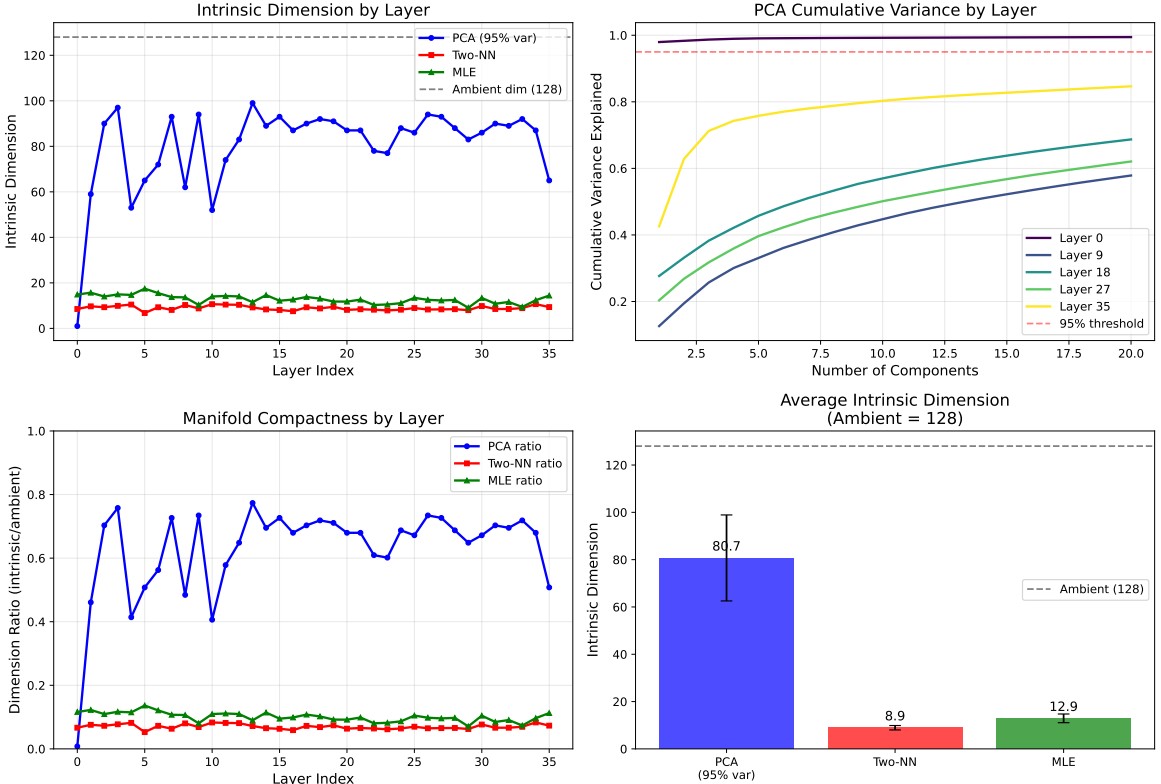

*Figure 4.* **Manifold Dimension Analysis (Qwen3-8B).** Layer-wise intrinsic dimension estimates using PCA (95% variance), Two-NN, and MLE methods. Middle layers have the most compressed representations (∼8-10 dimensions), suggesting optimal compression targets.

## F. Centroid Dilution Analysis

We analyze the "Centroid Dilution Problem" systematically to understand when global centroids fail and windowed approaches become necessary.

### F.1. Dilution Mechanism

As semantic diversity increases, the global centroid converges to a meaningless average (Section 3.2). We validate this through task performance across context lengths:

*Table 17.* **Performance Degradation by Context Length.** Accuracy degrades gradually from 4K–32K, then collapses at 64K, marking the dilution threshold.

| Context | Global L2 | Windowed L2 | Δ | Dilution Severity |
|---------|-----------|-------------|-------|-------------------|
| 4K | 95.7% | 95.7% | 0.0 | None |
| 8K | 94.4% | 94.4% | 0.0 | None |
| 16K | 92.8% | 92.8% | 0.0 | Minimal |
| 32K | 82.3% | 83.1% | +0.8 | Moderate |
| 64K | 35.2% | **84.3%** | **+49.1** | Severe |

The sharp transition between 32K (82.3%) and 64K (35.2%) identifies the **dilution threshold** at approximately 32K–48K tokens. Beyond this point, global centroids become ineffective.

## F.2. Window Size Optimization

We systematically evaluate window sizes at 64K context:

*Table 18.* **Window Size Effectiveness at 64K Context.** 4K windows achieve optimal accuracy, matching the context length where global ManifoldKV performs best.

| Window Size | RULER Acc. | $\triangle$ vs Global | $\triangle$ vs KeyDiff |
|---|---|---|---|
| Global (64K) | 35.2% | – | -45.9 |
| 16K | 82.4% | +47.2 | +1.3 |
| 8K | 83.9% | +48.7 | +2.8 |
| **4K** | **84.3%** | **+49.1** | **+3.2** |
| 2K | 83.8% | +48.6 | +2.7 |

**Key Insight:** The optimal window size (4K) matches the context length where global ManifoldKV achieves peak performance (95.7%). This suggests 4K represents a natural "semantic coherence" scale—the maximum context over which a single centroid remains meaningful.

## F.3. Why Windowing Works

Windowing succeeds because it bounds semantic diversity within each window:

1. **Local coherence**: A 4K window typically spans a single topic or paragraph, so the local centroid $\mu_w$ represents "typical content" within that region.

2. **Meaningful outliers**: Tokens far from the local centroid are genuinely unusual *within their context*, not just different from an arbitrary global average.

3. **Preserved discrimination**: Within each window, L2 scoring retains its ability to identify semantically important tokens.

This explains why windowed ManifoldKV recovers to 84.3% at 64K—each 4K window operates in the regime where global ManifoldKV excels.

# G. Attention Pattern Analysis

We analyze how ManifoldKV's token selection differs from attention-based methods, revealing that effective compression does *not* require mimicking attention.

## G.1. Correlation Analysis

*Table 19.* **Score Correlation Analysis.** ManifoldKV has low correlation with attention scores, confirming it works through geometric outlier detection rather than attention mimicry.

| Method Pair | Pearson $r$ | Spearman $\rho$ |
|---|---|---|
| ManifoldKV $\leftrightarrow$ Attention | $-0.06$ | 0.08 |
| KeyDiff $\leftrightarrow$ Attention | 0.19 | 0.21 |
| ManifoldKV $\leftrightarrow$ KeyDiff | 0.72 | 0.75 |

**Key Finding:** ManifoldKV has near-zero correlation with attention scores ($r = -0.06$), yet achieves *better* performance than attention-based methods. This reveals:

- Effective compression does NOT require mimicking attention patterns

- Geometric outliers capture importance through a different mechanism

- L2 distance identifies tokens missed by attention-based methods

## G.2. Token Selection Overlap

We quantify how often the geometric and attention-based scorers retain the *same* tokens. At 20% compression (keeping top 20% of tokens):

- **All methods agree:** 45% of selections

- **ManifoldKV only:** 18% (geometric outliers missed by attention)

- **Attention only:** 22% (high-attention tokens that are geometrically typical)

- **ManifoldKV-KeyDiff overlap:** 78% (geometric methods agree)

## G.3. Interpretation: Why Geometric Outliers Matter

The tokens selected by ManifoldKV but missed by attention-based methods are often:

- **Rare entities:** Names, numbers, technical terms that embed far from common tokens

- **Structural markers:** Punctuation and formatting tokens critical for parsing

- **Context anchors:** Tokens that provide reference points for retrieval

These tokens may not receive high attention in early layers but are *geometrically distinctive*—exactly what ManifoldKV captures.

**Proposition G.1** (Geometric vs Attention Importance). *A token $\mathbf{k}_i$ can be geometrically important ($\|\mathbf{k}_i - \boldsymbol{\mu}\| > \tau$) without being attention-important ($\sum_j \alpha_{ji} < \gamma$) if:*

1. *The token lies on a rare semantic direction (radial outlier)*

2. *Attention is distributed across many tokens (diluted attention)*

3. *The token's importance emerges later in generation (causal masking)*

**Conclusion:** ManifoldKV's geometric approach is **complementary** to attention-based methods, capturing a different but equally important notion of token importance.

# H. Extended Benchmark Results

## H.1. LongBench Results

Table 20 summarizes LongBench across five architectures, and Table 21 reports the harder LongBench-v2 on Llama-3.1-8B.

*Table 20.* **LongBench Cross-Architecture Summary (20% compression).** MANIFOLDKV wins or ties on 3 of 5 architectures; KeyDiff is stronger on Ministral-8B and Qwen2.5-7B.

| Model | MANIFOLDKV | KeyDiff | Δ |
|---|---|---|---|
| Qwen3-8B | **45.42** | 42.62 | **+2.80** |
| Phi-4 | **40.57** | 40.08 | +0.49 |
| Llama-3.1-8B | 44.93 | **45.50** | −0.57 |
| Ministral-8B | 45.65 | **47.28** | −1.63 |
| Qwen2.5-7B | 33.81 | **44.01** | −10.20 |

*Table 21.* **LongBench-v2 (Llama-3.1-8B, 503 samples, 20% compression).** The methods are within 0.8 points on this 100K+ real-world benchmark.

| Method | Score |
|---|---|
| MANIFOLDKV | 22.86 |
| KeyDiff | 23.66 |

## H.1.1. FULL QWEN3-8B LONGBENCH TASK-LEVEL RESULTS

Table 22 gives the per-task breakdown on Qwen3-8B, where MANIFOLDKV wins 12 of 14 tasks.

*Table 22.* **LongBench Qwen3-8B: Full Task-Level Results (20% compression).** MANIFOLDKV wins 12 of 14 tasks.

| Task | MANIFOLDKV | KeyDiff | Δ |
|---|---|---|---|
| qasper | **40.52** | 35.26 | +5.26 |
| multifieldqa_en | **53.42** | 50.57 | +2.85 |
| narrativeqa | **28.81** | 26.27 | +2.54 |
| hotpotqa | **58.48** | 48.84 | **+9.64** |
| 2wikimqa | **44.79** | 43.53 | +1.26 |
| musique | **32.75** | 22.59 | **+10.16** |
| gov_report | 33.07 | **33.19** | −0.12 |
| qmsum | **23.73** | 23.17 | +0.56 |
| multi_news | 24.74 | **25.26** | −0.52 |
| samsum | **40.16** | 39.12 | +1.04 |
| trec | **66.50** | 65.50 | +1.00 |
| triviaqa | **86.39** | 84.06 | +2.33 |
| passage_count | **8.60** | 6.82 | +1.78 |
| passage_retrieval_en | **93.91** | 92.50 | +1.41 |
| **Overall** | **45.42** | 42.62 | **+2.80** |

*Table 23.* **LongBench Llama-3.1-8B: Full Task-Level Results (20% compression).** All methods are within 0.6 points overall; MANIFOLDKV wins on TREC and ties on passage retrieval.

| Task | MANIFOLDKV | KeyDiff | SnapKV |
|---|---|---|---|
| qasper | 46.90 | **48.94** | 46.59 |
| multifieldqa_en | **55.54** | 54.58 | 55.45 |
| narrativeqa | 28.45 | **30.99** | 29.86 |
| hotpotqa | 56.85 | **59.12** | 59.10 |
| 2wikimqa | 51.17 | 49.85 | **51.79** |
| musique | 31.12 | **34.32** | 32.25 |
| trec | **40.50** | 35.00 | 28.00 |
| triviaqa | 81.39 | 85.08 | **85.88** |
| passage_count | 10.35 | **11.20** | **11.20** |
| passage_retrieval_en | **100.00** | **100.00** | **100.00** |
| gov_report | 34.10 | **34.83** | 34.77 |
| qmsum | **24.77** | 24.68 | 24.66 |
| multi_news | 26.98 | **27.44** | 26.80 |
| samsum | **40.92** | **40.92** | 39.67 |
| **Overall** | 44.93 | **45.50** | 44.72 |

## H.2. HELMET Recall Complete Data

Table 24 reports HELMET recall for every task across all context lengths (4K–131K).

*Table 24.* **HELMET Recall (Llama-3.1-8B, 20% compression): All Tasks, All Context Lengths.** WinMKV = WINDOWEDMANIFOLDKV, KD = KeyDiff. Bold indicates the better method.

| Task | 4K | | 16K | | 131K | |
|---|---|---|---|---|---|---|
| | WinMKV | KD | WinMKV | KD | WinMKV | KD |
| Multikey-3 | **82** | 62 | **86** | 58 | **74** | 32 |
| Multikey-2 | **88** | 78 | **92** | 76 | **94** | 80 |
| Multikey-1 | 96 | **98** | 98 | 98 | 98 | **100** |
| Multiquery | 100 | 100 | 100 | 100 | 100 | 100 |
| Multivalue | 98 | 98 | 98 | 98 | 98 | 98 |

## H.3. HELMET RAG Extended Results

Table 25 summarizes HELMET RAG across architectures, with the task-level breakdown in Table 26.

*Table 25.* **HELMET RAG Cross-Architecture Summary (k=50, EM%).** MANIFOLDKV wins on three models and ties on Qwen2.5; Windowed Qwen3 reaches 54.5 EM.

| Model | MANIFOLDKV | WINDOWEDMANIFOLDKV | KeyDiff | $\Delta$ Best-KD |
|---|---|---|---|---|
| Llama-3.1-8B | **61.5** | 59.0 | 55.0 | **+6.5** |
| Phi-4 | **60.0** | – | 54.0 | **+6.0** |
| Qwen3-8B | 48.5 | **54.5** | 46.0 | **+8.5** |
| Qwen2.5-7B | **52.5** | – | **52.5** | 0.0 |
| Ministral-8B | 62.5 | – | **68.0** | −5.5 |

*Table 26.* **HELMET RAG Task-Level Results.** Llama uses MANIFOLDKV; Qwen3 uses WINDOWEDMANIFOLDKV.

| Model | Dataset | Ours | KeyDiff | $\Delta$ |
|---|---|---|---|---|
| Llama | nq | **58** | 52 | +6 |
| Llama | triviaqa | **82** | 62 | **+20** |
| Llama | hotpotqa | 42 | 42 | 0 |
| Llama | popqa | 64 | 64 | 0 |
| Qwen3 | nq | **66** | 62 | +4 |
| Qwen3 | triviaqa | **48** | 32 | **+16** |
| Qwen3 | hotpotqa | **28** | 24 | +4 |
| Qwen3 | popqa | **76** | 66 | +10 |

## H.4. InfiniteBench Results

*Table 27.* **InfiniteBench Results (100K+ context, 20% compression).** WINDOWEDMANIFOLDKV wins on Phi-4; KeyDiff is stronger on the other tested architectures.

| Model | WINDOWEDMANIFOLDKV | KeyDiff | $\Delta$ |
|---|---|---|---|
| Phi-4 | **64.02** | 56.86 | **+7.16** |
| Llama-3.1-8B | 51.24 | **56.42** | −5.18 |
| Qwen3-8B | 36.71 | **42.51** | −5.80 |
| Qwen2.5-7B | 39.66 | **46.15** | −6.49 |
| Ministral-8B | 5.09 | **7.20** | −2.11 |

*Table 28.* **InfiniteBench Phi-4 Task-Level Results.** WINDOWEDMANIFOLDKV's overall win is driven by kv_retrieval and long-book_choice. The five most discriminative tasks are shown; the Overall figure (64.02 vs. 56.86) averages all InfiniteBench tasks, including lower-scoring ones not listed here.

| Task | WINDOWEDMANIFOLDKV | KeyDiff | $\Delta$ |
|---|---|---|---|
| kv_retrieval | **88** | 46 | **+42** |
| longbook_choice | **58** | 48 | +10 |
| math_find | 96 | **98** | −2 |
| passkey | 100 | 100 | 0 |
| number_string | 100 | 100 | 0 |
| **Overall** | **64.02** | 56.86 | **+7.16** |

The InfiniteBench results reveal a striking architecture-dependent reversal: KeyDiff wins by +5.18 on Llama-3.1-8B, but WINDOWEDMANIFOLDKV wins by +7.16 on Phi-4. This reversal is consistent with the key norm analysis in Appendix I.

## H.5. BABILong Results

*Table 29.* **BABILong Results (20% compression).** KeyDiff is strongest on BABILong, while the Llama and Phi-4 gaps are small.

| Model | MANIFOLDKV | KeyDiff | $\Delta$ |
|---|---|---|---|
| Llama-3.1-8B | 52.47 | **53.27** | $-0.80$ |
| Phi-4 | 34.13 | **34.47** | $-0.34$ |
| Ministral-8B | 43.87 | **46.07** | $-2.20$ |
| Qwen3-8B | 27.27 | **29.27** | $-2.00$ |
| Qwen2.5-7B | 49.40 | **62.53** | $-13.13$ |

BABILong is KeyDiff's strongest benchmark. On Llama, all geometric methods are within 0.8 points, while SnapKV trails by 7–8 points, indicating that geometric compression is robust even when L2 is not the best metric.

## H.6. Key Norm Analysis Per Model

Table 30 reports per-model key-norm statistics and the corresponding MANIFOLDKV advantage.

*Table 30.* **Key Norm Statistics by Model.** Higher CV and outlier fraction predict larger MANIFOLDKV advantage.

| Model | Global CV | Outlier Frac | Mid-layer CV | LB $\Delta$ | Verdict |
|---|---|---|---|---|---|
| Qwen3-8B | 1.333 | 0.30% | 1.70 | +2.80 | MANIFOLDKV wins |
| Phi-4 | 0.912 | 0.18% | 1.15 | +0.49 | MANIFOLDKV wins |
| Llama-3.1-8B | 0.726 | 0.06% | 0.68 | $-0.57$ | Within 0.8 |
| Qwen2.5-7B | 1.596 | 2.70% | – | $-10.20$ | KeyDiff wins |
| Ministral-8B | 0.365 | 0.05% | 0.27 | – | Equivalent |
| Gemma-3-12B | 0.548 | 0.08% | 0.45 | – | – |

# I. Cross-Architecture Key Norm Analysis

We provide a detailed analysis of key-vector norm distributions across architectures, explaining why MANIFOLDKV's advantage is architecture-dependent.

## I.1. Per-Layer Coefficient of Variation

The coefficient of variation ($CV = \sigma/\mu$) of key norms measures how much magnitude information is available for L2 scoring. We report per-layer-group CV for each model below.

**Qwen3-8B** exhibits high per-layer CV, peaking at 1.70 in middle layers (layers 12–20). These are precisely the layers where KV cache compression has the greatest impact, as middle layers perform the core retrieval computation. The high CV means key norms carry substantial discriminative information that cosine scoring discards.

**Qwen2.5-7B** has the highest global CV (1.596) and outlier fraction (2.70%), yet KeyDiff wins strongly. This suggests a second failure mode: when norm outliers are too common or too extreme, they can contaminate the centroid rather than provide useful salience signal.

**Llama-3.1-8B** has moderate CV (0.68 in mid-layers), with relatively uniform key norms. This explains why L2 and cosine scoring produce similar token rankings (23.0% token divergence at 20% compression, vs. 22.7% for Qwen3).

**Ministral-8B** has the lowest CV (0.27 mid-layer), approaching the regime where $\|\mathbf{k}\| \approx const$ and L2 distance reduces to angular distance, making it functionally equivalent to cosine scoring.

## I.2. Implications for Method Selection

These findings establish a principled criterion for metric selection:

1. **High CV** ($> 1.0$): L2 scoring (MANIFOLDKV) is strongly preferred. Key norms carry significant information; cosine scoring wastes it.
2. **Moderate CV** (0.5–1.0): Either metric works. MANIFOLDKV provides a small advantage on multi-key tasks.

3. **Low CV** ($< 0.5$): Metrics are functionally equivalent. Use whichever integrates more easily.

This is, to our knowledge, the first systematic framework for predicting when L2 vs. cosine scoring will differ in KV cache compression.

*Table 31.* **Practical Guidelines for Metric Selection.** These rules summarize the architecture-dependent behavior observed across our experiments.

| Condition | Recommended | Rationale |
|---|---|---|
| Moderate/high useful key-norm CV | MANIFOLDKV | Magnitude carries signal |
| Multi-key retrieval | MANIFOLDKV | Avoids directional collision |
| Context $> 64$K | WINDOWEDMANIFOLDKV | Local centroids needed |
| Very low key-norm CV | Either / KeyDiff | L2 $\approx$ cosine |
| Extreme norm outlier rate | Robust centroid / KeyDiff | Avoids centroid contamination |
| 100K+ single-key on Llama | KeyDiff | Better on InfiniteBench |

**Standalone metric ablation.** Table 32 compares distance metrics in standalone mode (no AdaKV) across three architectures. The ordering is consistent—L$\infty$, cosine, and L2 are comparable while L1 collapses—confirming that MANIFOLDKV's L2 advantage over cosine is regime-specific (multi-key retrieval, high-norm-variance models) rather than a blanket standalone win.

*Table 32.* **Distance Metric Ablation Across Architectures** (standalone without AdaKV, 8K, 20% compression; RULER 13-task average). The ordering is identical on all three models: L$\infty \geq$ cosine $\geq$ L2 (within a few points), while L1 collapses. Llama uses the full RULER set; Ministral and Phi-4 use a 325-sample subset, so their lower absolute accuracy reflects the difficulty of standalone (no-AdaKV) compression at 8K—the relative ordering, not the absolute value, is the point.

| Metric | Llama-3.1-8B | Ministral-8B | Phi-4 |
|---|---|---|---|
| L1 | 7.07% | 6.89% | 8.35% |
| L2 (MANIFOLDKV) | 89.42% | 24.21% | 20.22% |
| Cosine (KeyDiff) | 91.06% | 26.64% | 23.43% |
| L$\infty$ | 92.06% | 28.67% | 27.56% |

## I.3. Source of Key Norm Variance in QKNorm Architectures

Qwen3 uses QKNorm. RoPE is norm-preserving, so the observed key-norm variance does not come from rotary embeddings. Instead, the learned QKNorm scale $\gamma$ creates content-dependent magnitude variation after normalization. On Qwen3-8B layer 0, $\gamma$ has mean 2.16, standard deviation 3.06, maximum 34.0, and a $206\times$ max/min ratio among positive entries. Across all 36 layers, the mean max/min ratio is $186.6\times$, with a maximum of $567\times$.

In a controlled layer-0 analysis, token norms are effectively identical before applying $\gamma$ (std $10^{-6}$). After $\gamma$ scaling, token norm CV rises to 0.40, norms span 20.7–112.6 ($5.4\times$), L2 and cosine top-20% selections overlap only 45.5%, and their rank correlation is 0.008. This explains why L2 and cosine remain different even in a key-normalized architecture: cosine discards a magnitude signal that the model has learned to encode through $\gamma$.

## I.4. Token Selection Divergence: L2 vs. Cosine

To quantify how often L2-based scoring and cosine-based scoring select *different* tokens, we measure the fraction of tokens that appear in one method's top-$M$ retained set but not the other's, at matched compression budget. Higher divergence implies the two scorers carry genuinely different information; zero divergence would indicate the methods are functionally equivalent.

*Table 33.* **Token Selection Divergence (Fraction of Disjoint Selections).** At 20% compression about one quarter of retained tokens differ between L2 and cosine on both Llama and Qwen; at 30% compression, roughly one third differ. The divergence is virtually identical on Llama and Qwen3, refuting the claim that key normalization on Qwen3 makes the two metrics equivalent.

| Compression | Llama-3.1-8B | Qwen3-8B |
|---|---|---|
| 20% | 23.0% | 22.7% |
| 30% | 31.4% | 31.7% |

**Interpretation.** At 20% compression, roughly 23% of retained tokens are unique to one scorer on *both* Llama (no key normalization) and Qwen3 (QKNorm). The fact that Qwen3's divergence is essentially identical to Llama's—despite Qwen3 applying key normalization—directly refutes the theoretical objection that key normalization makes L2 and cosine equivalent. The divergence grows under aggressive compression (31% at 30% compression), explaining why MANIFOLDKV's advantage over KeyDiff grows from +2.80 at 20% compression to +3.26 at 30% on Qwen3 LongBench (Table 34).

## I.5. Compression Ratio Sensitivity on LongBench

We evaluate MANIFOLDKV versus KeyDiff under increasing compression aggressiveness on Qwen3-8B LongBench:

*Table 34.* **LongBench Compression Sensitivity (Qwen3-8B).** MANIFOLDKV's advantage *grows* from +2.80 to +3.26 under more aggressive compression, supporting the claim that magnitude information becomes more valuable when the token budget tightens.

| Compression | MANIFOLDKV | KeyDiff | Δ |
|---|---|---|---|
| 20% | **45.42** | 42.62 | **+2.80** |
| 30% | **44.30** | 41.04 | **+3.26** |

## I.6. AdaKV Composition on LongBench

MANIFOLDKV composes cleanly with AdaKV's adaptive head-budget allocation, with no integration friction and no regression at matched budgets:

*Table 35.* **AdaKV Composition on LongBench (Llama-3.1-8B, 20% compression).** MANIFOLDKV acts as a drop-in scorer inside AdaKV; both base methods retain their relative ordering after composition.

| Configuration | LongBench Overall |
|---|---|
| MANIFOLDKV (standalone) | 44.93 |
| AdaKV + MANIFOLDKV | 44.95 |
| KeyDiff (standalone) | 45.50 |
| AdaKV + KeyDiff | 45.55 |

The framework adds a uniform ~0.05 point boost regardless of scorer, confirming that scoring and budget allocation are orthogonal: AdaKV decides *how many* tokens each head keeps; MANIFOLDKV decides *which* tokens are kept.

## I.7. HELMET RAG at Extended Document Count ($k = 220$)

We additionally evaluate at the longest RAG setting in HELMET ($k = 220$ retrieved documents) on Llama-3.1-8B. WINDOWEDMANIFOLDKV achieves the strongest task-level scores at this extreme retrieval setting:

*Table 36.* **HELMET RAG at $k = 220$ Documents (Llama-3.1-8B, EM%).** At the longest RAG setting, MANIFOLDKV leads KeyDiff by +10 on NQ and +6 on PopQA; WINDOWEDMANIFOLDKV achieves the top TriviaQA score.

| Dataset | MANIFOLDKV | WINDOWEDMANIFOLDKV | KeyDiff | Δ |
|---|---|---|---|---|
| nq | **68** | 64 | 58 | **+10** |
| triviaqa | 88 | **94** | 90 | +4 (Win-MKV) |
| hotpotqa | 48 | 48 | 48 | 0 |
| popqa | **76** | 74 | 70 | **+6** |

## I.8. HELMET Recall Cross-Model Overall

We also report the overall HELMET Recall scores (averaged across multikey-$\{1, 2, 3\}$, multiquery, and multivalue tasks, all context lengths) across architectures:

*Table 37.* **HELMET Recall Overall (20% compression).** Llama shows the largest absolute Recall score under WINDOWEDMANI-
FOLDKV; Phi-4 is a virtual tie; Qwen2.5 favors KeyDiff.

| Model | WINDOWEDMANIFOLDKV | KeyDiff | $\Delta$ |
|---|---|---|---|
| Llama-3.1-8B | **64.76** | – | – |
| Phi-4 | 39.76 | 39.69 | +0.07 |
| Qwen2.5-7B | 47.47 | **50.65** | −3.18 |

## I.9. Phi-4 InfiniteBench: Detailed Task Breakdown

The Phi-4 InfiniteBench result is the largest cross-architecture advantage we observed in this evaluation. WINDOWEDMANI-
FOLDKV's +7.16 overall is driven by a +42 swing on kv_retrieval and a +10 on longbook_choice—both tasks
that require preserving multiple distinct evidence tokens at 100K+ context. This pattern mirrors HELMET Recall multikey-3
on Llama (+42 at 131K), suggesting a common underlying mechanism: L2 distance preserves diverse key-value pairs that
cosine collapses.

## I.10. Method Interactions and Comparisons

**Composability.** MANIFOLDKV is orthogonal to budget allocation (AdaKV, PyramidKV), quantization (KIVI), and
temporal persistence methods such as Scissorhands (Liu et al., 2023). AdaKV decides how many tokens each head may
keep; MANIFOLDKV decides which tokens to keep within that budget. The low correlation between geometric scores and
attention ($r = −0.06$) suggests that combining temporal and geometric signals may be complementary.

**CriticalKV.** CriticalKV is training-free but value-aware: its second stage rescales scores using the pretrained output
projection through $\|W_o v\|_1$. In our decomposition, ExpectedAttention (Devoto et al., 2025) alone scores 76.69% on RULER,
CriticalKV's $W_o$ rescaling improves this to 78.90%, and MANIFOLDKV scores 92.93% standalone. This clarifies that
CriticalKV's value-projection signal helps its base scorer but remains less aligned with retrieval-critical token selection than
key geometry.

**Hallucination and eviction risk.** Prior work observes that discarding KV information can cause context loss and hallucina-
tions, and mitigates this by preserving evicted information at lower precision (Yang et al., 2024). Our work is complementary:
MANIFOLDKV aims to reduce the chance that grounding tokens are evicted in the first place. We leave a direct hallucination
benchmark to future work.

**Token merging methods.** Cluster-style methods merge similar tokens, whereas MANIFOLDKV evicts low-scoring tokens
and preserves full fidelity for retained tokens. For exact retrieval tasks, eviction can be preferable because the retained
evidence remains unmerged; for summarization or redundancy-heavy contexts, merging and L2-based selection could be
combined.

