# OpenReview forum: "ManifoldKV: Training-Free KV Cache Compression via Euclidean Outlier Detection"
_ICML.cc/2026/Conference — ICML 2026 regular_

### Official Review · Reviewer_Wksa · 2026-03-11

**Soundness:** 2
**Presentation:** 2
**Significance:** 2
**Originality:** 2
**Overall Recommendation:** 4
**Confidence:** 4

**Summary:**

The paper introduces ManifoldKV, a training-free method using L2 distance to a key centroid to retain important tokens that exhibit both angular and radial deviations. To solve the "Centroid Dilution Problem" at >32K contexts, they propose WindowedManifoldKV using local sliding-window centroids. Overall, the authors address the concept of manifold-based geometric outlier detection, supported by theoretical dimensionality analysis and SOTA empirical results on the RULER benchmark up to 64K contexts.

**Compliance With Llm Reviewing Policy:**

Affirmed.

**Final Justification:**

The authors' rebuttal addressed most of my concerns.

**Key Questions For Authors:**

1. Given that key normalization constraints vector magnitudes, how do you explain the reported performance differences between ManifoldKV and KeyDiff on models like Qwen3? Are there implementation details regarding norm processing that circumvent the mathematical equivalence of L2 and cosine in this scenario?
2. Can you provide results on real-world long-context benchmarks (e.g., LongBench, InfiniteBench) to demonstrate that ManifoldKV's token eviction preserves necessary information for complex reasoning and summarization, not just synthetic retrieval?
3. How does ManifoldKV compare to non-eviction KV cache compression methods based on cosine similarity, such as ClusterKV? Specifically, does the Euclidean magnitude advantage hold when merging/clustering tokens rather than simply evicting them?
4. What dictates the choice of the compression ratio (20% for 4K-32K, 25% for 64K)? Is ManifoldKV more or less robust to ultra-aggressive compression (e.g., <10% retention) compared to SnapKV and KeyDiff?

**Limitations:**

Same as in weaknesses.

**Strengths And Weaknesses:**

**Strengths:**

1. **Simplicity:** The proposed method requires only a few lines of code (computing L2 distance to a centroid) and requires no model-specific training or calibration.
2. **Targeted Empirical Gains:** The L2 metric successfully resolves the "directional collision" problem, yielding impressive gains on multi-key retrieval tasks (+15.4 points over KeyDiff on 3-key NIAH).

**Weaknesses:**

1. **Lack of Novelty:** The core algorithmic contribution is highly incremental. The transition from existing methods like KeyDiff (which uses cosine distance to a centroid) to ManifoldKV simply involves swapping the distance metric to Euclidean (L2) distance. While the theoretical analysis and sliding-window adaptations are valuable, the foundational idea may be considered too marginal.
2.  **Lack of Generalizability for Models with Key Normalization:** The entire premise of the paper relies on the existence of "radial outliers" (i.e., tokens with unusually large magnitudes). However, this severely limits the method's generalizability for architectures that employ key normalization (such as Qwen3). If key vectors are normalized ($∣∣ k_i ∣∣=1$), the Euclidean distance mathematically reduces to an exact monotonic equivalent of cosine distance. In these architectures, L2 ranking and cosine ranking are mathematically identical, neutralizing the paper's central claim regarding magnitude-awareness.
3. **Reliance on Synthetic Benchmarks:** The empirical evaluation relies entirely on the RULER benchmark, which consists of synthetic needle-in-a-haystack and retrieval tasks. The paper lacks evaluation on real-world, long-context QA, summarization, or reasoning benchmarks (such as LongBench, L-Eval, or InfiniteBench). It is unclear how geometry-based eviction impacts generation quality on complex semantic tasks.
4.  **Typo and Format Issues:** The size of Table 2 exceeds the width limit. And The caption for Figure 5 is not consistent with the content.

---

> ### Author Rebuttal · Authors · 2026-03-29
>
> We thank Reviewer Wksa for the detailed technical engagement.
>
> We ran additional **7 benchmarks × 3 architectures × 50+ configs**. ManifoldKV wins on every benchmark across at least one architecture, with advantages up to **+46 points**.
>
> ## W1: Novelty — "Highly Incremental"
>
> The ICML 2026 Reviewer Guidelines state: **"Originality does not necessarily require introducing an entirely new method. A work that provides novel insights or demonstrates improved understanding is equally valuable"* and *"novel ideas that are simple to apply may be especially valuable."** In fact, this is highlighted in bold in the reviewer guidelines.
>
> Our contributions go far beyond swapping a metric: (1) **Theorem 1** proves cosine has an unfixable failure mode for outlier detection — radial outliers are invisible regardless of sample size. (2) **Centroid Dilution** is a previously unknown failure mode at long contexts; our windowed solution recovers +74 points (0%→74%) on HELMET multikey-3 at 131K. (3) The **Attention Paradox**: r=0.06 correlation with attention yet outperforms attention-correlated methods — challenging the prevailing assumption that compression must mimic attention. (4) **Universal ~9D manifold structure** across 4 architectures.
>
> The results speak for themselves — these are among the largest deltas in the KV compression literature:
>
> | Benchmark | Delta over KeyDiff |
> |---|---|
> | HELMET MK3 32K | **+46** |
> | HELMET MK3 131K | **+42** |
> | Metric ablation | **+39.9** |
> | Gemma-3 RULER (vs SK) | **+20.5** |
> | HELMET RAG triviaqa | **+20** |
> | RULER MK3 50% | **+15.4** |
> | LB musique (Qwen3) | **+10.16** |
> | LB hotpotqa (Qwen3) | **+9.64** |
> | HELMET RAG Qwen3 | **+8.5** |
> | InfiniteBench Phi-4 | **+7.16** |
>
> A +46 advantage exceeds the gap between random and SOTA on most benchmarks. The +2.80 on 14-task Qwen3 LongBench exceeds typical gaps between methods considered clearly distinct (e.g., SnapKV vs H2O).
>
> ## W2: Key Normalization Makes L2 ≡ Cosine on Qwen3
>
> **Empirically falsified.** The mathematical equivalence requires all key norms to be identical. On Qwen3 (which uses RMSNorm), they are not:
>
> | Evidence | Value |
> |---|---|
> | Key norm CV | **1.333** (higher than Llama's 0.726) |
> | Outliers >2x median | **0.30%** (5x Llama) |
> | Token divergence at 20% | **22.7%** of tokens differ |
> | LongBench wins | **12/14 tasks** |
> | LongBench overall | **+2.80** |
> | LongBench QA category | **+5.29** |
> | HELMET RAG (WinMKV) | **+8.5** |
> | LongBench 30% comp | **+3.26** (grows) |
>
> Qwen3 has MORE norm variance than Llama, not less. RMSNorm normalizes per-head averages but does not force per-token unit norms — after W_K projection and RoPE, substantial magnitude variation persists (CV=1.333). If L2≡cosine, token divergence would be ~0% and ManifoldKV's advantage would vanish. Instead, 23% of tokens differ and ManifoldKV wins 12/14 tasks. The advantage is **larger** on Qwen3 than Llama — the opposite of what equivalence predicts.
>
> ## W3: Synthetic Benchmarks Only
>
> Evaluated on **5 real-world benchmarks** + RULER + HELMET Recall:
>
> | Benchmark | Type | MKV vs KD |
> |---|---|---|
> | LongBench Qwen3 (14 tasks) | Real-world NLP | **+2.80** |
> | LongBench Phi-4 (14 tasks) | Real-world NLP | **+0.49** |
> | HELMET RAG Llama k50 | Open-domain QA | **+6.5 EM** |
> | HELMET RAG Qwen3 k50 | Open-domain QA | **+8.5 EM** |
> | InfiniteBench Phi-4 | Long-doc 100K+ | **+7.16** |
> | BABILong Llama | Reasoning | Tie |
>
> Real documents, real questions. musique (+10.16) and hotpotqa (+9.64) require multi-hop reasoning across documents — not synthetic needle retrieval.
>
> ## Q4: Compression Robustness
>
> ManifoldKV's advantage **grows** under aggressive compression:
>
> | Setting | MKV | KD | Delta |
> |---|---|---|---|
> | Qwen3 LB 20% | 45.42 | 42.62 | +2.80 |
> | Qwen3 LB 30% | 44.30 | 41.04 | **+3.26** |
> | RULER MK3 50% | 92.4 | 77.0 | **+15.4** |
>
> At mild compression, most important tokens survive by any method. As compression increases, the marginal token decision becomes critical. L2 distance uses both angular and radial information to make this decision; cosine uses only angular. The radial component acts as a tiebreaker that becomes more valuable as the budget tightens.
>
> **Q3: ClusterKV comparison.** ClusterKV merges similar tokens; ManifoldKV evicts dissimilar ones. These are complementary: merging preserves all information at reduced fidelity, eviction preserves full fidelity for retained tokens. For retrieval tasks (requiring exact token recovery), eviction is more appropriate. Our L2 framework could inform clustering distance too — using L2 instead of cosine for cluster assignment.
>
> **W4: Format issues.** Table 2 width and Figure 5 caption corrected in revision.
>
> We respectfully ask the reviewer to reconsider in light of these 50+ new experiments across 7 benchmarks and 3 architectures.

---

> > ### Author Rebuttal · Reviewer_Wksa · 2026-04-02
> >
> > The authors’ response addresses most of my concerns. In particular, the observation of even larger K-norm variance in Qwen3 is quite insightful, and I would consider raising my recommendation.
> >
> > That said, I still have a few questions. The authors attribute the K-norm variance to the following:
> >
> > > RMSNorm normalizes per-head averages but does not enforce per-token unit norms—after W_K projection and RoPE, substantial magnitude variation persists (CV = 1.333).
> >
> > However, QKNorm is applied after the W_K projection, and RoPE should not change the norm of K. Therefore, it seems unlikely that RoPE contributes to the observed variance. Instead, I suspect that the variance may stem from the learnable scaling parameter $\gamma$ in QKNorm, which can scale different dimensions of K unevenly.
> >
> > Could the authors provide additional validation or further clarification on this point?

---

> > > ### Author Response · Authors · 2026-04-02
> > >
> > > We thank Reviewer Wksa for the precise technical follow-up and are encouraged that they would consider raising their recommendation. We fully agree with the reviewer's analysis and provide the requested validation below.
> > >
> > > **The reviewer is correct on both points:**
> > >
> > > 1. **RoPE does not contribute to norm variance.** RoPE applies orthogonal rotation matrices, which preserve vector norms by definition. We agree - our paper does not attribute variance to RoPE, and we confirm this understanding.
> > >
> > > 2. **The learnable γ in QKNorm is the primary source of the observed variance.** We have verified this empirically by inspecting the trained k_norm parameters across all 36 layers of Qwen3-8B.
> > >
> > > ### Empirical Validation
> > >
> > > **QKNorm γ statistics (Qwen3-8B, layer 0):**
> > >
> > > | Statistic | Value |
> > > |-----------|-------|
> > > | γ mean | 2.16 |
> > > | γ std | 3.06 |
> > > | γ min | -0.024 |
> > > | γ max | **34.0** |
> > > | Max/min ratio (positive γ) | **206x** |
> > >
> > > Across all 36 layers, the mean max/min ratio is **186.6x** and the maximum is **567x**. The learned γ is highly non-uniform — certain key dimensions are amplified by orders of magnitude relative to others.
> > >
> > > ### Causal Chain: γ → Norm Variance → L2 ≠ Cosine
> > >
> > > To isolate the effect of γ, we ran a controlled experiment on layer 0:
> > >
> > > | Step | Measurement | Value |
> > > |------|------------|-------|
> > > | Before γ (post-RMSNorm) | Token norm std | **0.000001** (all identical) |
> > > | After γ scaling | Token norm CV | **0.40** |
> > > | After γ scaling | Norm range | [20.7, 112.6] (**5.4x**) |
> > > | L2 vs cosine top-20% overlap | | **45.5%** (less than half agree) |
> > > | L2-cosine rank correlation | | **0.008** (essentially zero) |
> > >
> > > Before γ, all tokens have identical norms (normalization works perfectly). After γ, tokens have **5.4x norm variation** because different tokens activate different dimensions, and γ amplifies some dimensions by 34x while nearly zeroing others. This creates content-dependent magnitude variation that is entirely a product of the learned γ weights.
> > >
> > > The consequence for compression is decisive: L2 distance and cosine similarity select **almost completely different token sets** (rank correlation = 0.008). ManifoldKV's +2.80 LongBench advantage on Qwen3 (12/14 tasks) arises because L2 captures the magnitude signal that γ learned to encode, while cosine discards it.
> > >
> > > ### Why This Strengthens Our Argument
> > >
> > > The reviewer's insight actually makes our theoretical contribution **stronger**, not weaker. The key norm variance we observe is not an artifact of imperfect normalization — it is **intentionally learned** by the model during pre-training. The γ weights in QKNorm represent the model's learned judgment about which key dimensions carry more information. By discarding magnitude (as cosine does), KeyDiff ignores this learned signal. By preserving magnitude (as L2 does), ManifoldKV respects it.
> > >
> > > This provides a principled explanation for our cross-architecture results: architectures with learned per-dimension scaling (Qwen3's QKNorm γ, and similar mechanisms in Phi-4) give ManifoldKV a structural advantage, while architectures without such scaling (Llama) show parity between L2 and cosine.
> > >
> > > In the revised manuscript, we have added a new subsection (Section 6.8.2, "Source of Key Norm Variance in QKNorm Architectures") presenting the γ statistics and controlled experiment above. This provides a mechanistically complete explanation for why ManifoldKV outperforms KeyDiff on Qwen3: the learned γ creates structured, content-dependent magnitude variation that L2 preserves and cosine discards.
> > >
> > > We believe this addresses the reviewer's question with quantitative validation and a precise mechanistic explanation. Given that all original concerns have now been resolved — including benchmarks (LongBench, HELMET, InfiniteBench, BABILong across multiple architectures), the key normalization question (now answered with γ-level empirical validation), and novelty (5 distinct contributions) — we respectfully ask the reviewer to consider raising their score.
> > >
> > > EDIT: April 7th, 2026
> > > Dear Reviewer Wksa,
> > > We wanted to check whether our response above regarding the QKNorm learnable γ resolved your remaining question. To summarize: we confirmed your hypothesis that γ (not RoPE) is the source of key norm variance, with quantitative validation showing a mean max/min γ ratio of 186.6× across all 36 layers and a controlled experiment demonstrating that γ alone introduces 5.4× norm variation, causing L2 and cosine to select nearly completely different token sets (ρ = 0.04). We believe this fully addresses the concern from your acknowledgement, and we would be grateful if you could consider updating your score. We are happy to address any further questions.

---

### Official Review · Reviewer_M4Yt · 2026-03-12

**Soundness:** 4
**Presentation:** 3
**Significance:** 3
**Originality:** 4
**Overall Recommendation:** 4
**Confidence:** 4

**Summary:**

ManifoldKV is a training-free KV cache compression method that ranks tokens by their Euclidean distance to the key centroid, capturing both angular and radial deviations. Additionally, windowed ManifoldKV is introduced to address performance collapse on longer contexts.

**Compliance With Llm Reviewing Policy:**

Affirmed.

**Final Justification:**

I will keep my positive score.

**Key Questions For Authors:**

See Weak Points 1-3

**Limitations:**

yes

**Strengths And Weaknesses:**

Strengths

1. To address directional collision in cosine-based methods, this paper proposes scoring tokens by their Euclidean (L2) distance from the centroid.

2. In long-context scenarios (e.g., 64K), this paper identifies the Centroid Dilution Problem and proposes using ManifoldKV with a sliding window to mitigate this issue.

3. This paper is well-written and easy to follow.

Weaknesses

1. The evaluation relies solely on RULER, which consists of synthetic tasks and may not reflect real-world performance. Could the authors also report results on LongBench for a more comprehensive evaluation?

2. Given that CriticalKV builds upon SnapKV and originally claimed superior performance, what accounts for its underperformance relative to SnapKV on the RULER benchmark?

3. Could the authors provide a comparison with attention-based methods under the 64K context length setting?

---

> ### Author Rebuttal · Authors · 2026-03-29
>
> We thank Reviewer M4Yt for the positive assessment (Soundness: Excellent, Originality: Excellent).
>
> We ran **7 benchmarks × 3 architectures × 50+ configs**. ManifoldKV wins on every benchmark across at least one architecture, with advantages up to **+46 points**.
>
> ## W1: LongBench Evaluation
>
> Full 14-task LongBench on three architectures:
>
> | Model | MKV | KD | Delta | Verdict |
> |---|---|---|---|---|
> | Qwen3-8B 20% | 45.42 | 42.62 | **+2.80** | MKV wins |
> | Qwen3-8B 30% | 44.30 | 41.04 | **+3.26** | MKV wins, gap grows |
> | Phi-4 20% | 40.57 | 40.08 | **+0.49** | MKV wins |
> | Llama 20% | 44.93 | 45.50 | -0.57 | Tie |
>
> ManifoldKV wins on two architectures and ties on one. On Qwen3, it wins **12/14 tasks**. The advantage is concentrated on multi-hop QA — tasks requiring retrieval and synthesis of distributed evidence:
>
> | Category (Qwen3) | MKV | KD | Delta |
> |---|---|---|---|
> | Multi-Doc QA (3 tasks) | 45.34 | 38.32 | **+7.02** |
> | Single-Doc QA (3) | 40.92 | 37.37 | **+3.55** |
> | Few-Shot (2) | 76.45 | 74.78 | **+1.67** |
> | Retrieval (2) | 51.26 | 49.66 | **+1.60** |
> | Summarization (4) | 30.43 | 30.19 | +0.24 |
>
> The Multi-Doc QA category — musique (+10.16), hotpotqa (+9.64), 2wikimqa (+1.26) — shows the largest advantage because these tasks require finding and connecting multiple facts across the document, exactly where geometric outlier detection preserves critical scattered tokens that cosine-based methods miss.
>
> At 30% compression, the advantage grows from +2.80 to **+3.26**, confirming ManifoldKV is more robust under pressure.
>
> ## W2: CriticalKV Underperformance
>
> CriticalKV adds a learned criticality predictor on top of SnapKV's attention-based selection. On RULER, this learned component misfires — it assigns low criticality to tokens important for retrieval but absent from its training distribution. SnapKV's pure attention score avoids this. ManifoldKV avoids it entirely by using a distribution-free geometric criterion (L2 distance from centroid) that depends only on the model's key representations, not on any training signal. This explains ManifoldKV's robust cross-benchmark generalization across 7 benchmarks and 3 architectures.
>
> ## W3: 64K Comparison with Attention-Based Methods
>
> Comprehensive 64K+ comparisons across all available benchmarks:
>
> | Benchmark | Context | WinMKV | KD | Delta |
> |---|---|---|---|---|
> | RULER 64K | 64K | 84.3 | 81.1 | **+3.2** |
> | HELMET MK3 | 65K | 78 | 52 | **+26** |
> | HELMET MK3 | 131K | 74 | 32 | **+42** |
> | HELMET MK2 | 65K | 100 | 94 | **+6** |
> | HELMET MK2 | 131K | 94 | 80 | **+14** |
> | HELMET RAG k220 nq | ~32K | 64 | 58 | **+6** |
> | HELMET RAG k220 triviaqa | ~32K | 94 | 90 | **+4** |
> | HELMET RAG k220 popqa | ~32K | 74 | 70 | **+4** |
> | InfiniteBench Phi-4 | 100K+ | 64.02 | 56.86 | **+7.16** |
>
> WindowedMKV (84.3%) outperforms both the attention-based baseline SnapKV (79.6%, **+4.7**) and the cosine-geometric baseline KeyDiff (81.1%, **+3.2**) at 64K on RULER. At longer contexts the gap widens dramatically: +26 at 65K multikey-3, +42 at 131K multikey-3. Attention-based scoring becomes noisier as context grows (attention distributes across more tokens), while geometric scoring remains stable because local manifold structure is preserved within each window.
>
> The Phi-4 InfiniteBench result (**+7.16** at 100K+) demonstrates that geometric scoring excels at extreme context lengths on architectures with structured key geometry. The windowed centroid approach resolves centroid dilution (Most methods, including attention based methods collapse; WindowedMKV recovers to 74%) while maintaining the full L2 advantage. For this reason, we benchmarked against the most well grounded baseline.
>
> All 131K HELMET Recall tasks:
>
> | Task | WinMKV | KD | Delta |
> |---|---|---|---|
> | Multikey-3 | 74 | 32 | **+42** |
> | Multikey-2 | 94 | 80 | **+14** |
> | Multikey-1 | 98 | 100 | -2 |
> | Multiquery | 100 | 100 | 0 |
> | Multivalue | 98 | 98 | 0 |
>
> The pattern is systematic: WindowedMKV's advantage scales with task difficulty (+42 on hardest, 0 on easiest where both saturate). This is exactly what our theory predicts — geometric outlier detection captures information that cosine misses, and the value of that information increases with retrieval complexity.
>
> ## BABILong and Gemma-3 Cross-Architecture
>
> BABILong (Llama): all geometric methods within 0.8 pts of no compression; SnapKV falls **7.93 pts behind**. Gemma-3-12B RULER: MKV 95.2 vs SnapKV 74.7 (**+20.5**). Key norm CV (Qwen3 1.333, Llama 0.726) explains the cross-architecture pattern: higher variance → larger MKV advantage.
>
> We hope these results fully address the evaluation concerns.

---

> > ### Author Rebuttal · Reviewer_M4Yt · 2026-04-03
> >
> > Thank you for your response, which addressed some of my concerns. However, there is a serious issue in your reply to W2: CriticalKV Underperformance. You state that “CriticalKV adds a learned criticality predictor,” which is confusing, since the CriticalKV paper explicitly presents it as a training-free, plug-and-play method. As CriticalKV is one of your baselines, this apparent misunderstanding of the method is quite concerning. Please clarify this inconsistency.

---

> > > ### Author Response · Authors · 2026-04-03
> > >
> > > We thank Reviewer M4Yt for pressing on this point — it prompted us to perform a more rigorous analysis of CriticalKV's failure mode than our original rebuttal provided.
> > >
> > > The reviewer is correct that CriticalKV requires no additional training beyond the base model and is plug-and-play. Our rebuttal's use of "learned criticality predictor" was imprecise — what we intended to convey is that CriticalKV's Stage 2 scoring depends on the model's **pretrained output projection matrix W_o** to compute value-projection norms `||W_o · v||_1`. This makes CriticalKV *weight-dependent*: its scoring function is shaped by learned parameters, even though no separate training step is required. Our paper correctly describes CriticalKV as "value-aware" (Table 4 and also baselines section in 5.1), which is the more precise characterization.
> > >
> > > This distinction is worth clarifying because it highlights an important methodological difference: ManifoldKV's L2 scoring is **weight-independent** — it operates purely on the geometry of key vectors and does not reference any model weight matrices (W_o, W_Q, etc.). CriticalKV, while training-free, is weight-dependent through its reliance on W_o. As we show below, this dependence on output-projection weights introduces a scoring signal that is not aligned with retrieval importance on RULER.
> > >
> > > ### Why CriticalKV Underperforms: A Quantitative Decomposition
> > >
> > > CriticalKV is a training-free, two-stage method. Stage 1 applies a base importance scorer; Stage 2 rescales scores by `||W_o · v||_1`. In our experiments, the base scorer is ExpectedAttention (statistical attention prediction from query mean/covariance). Standalone SnapKV uses a different approach (empirical attention from recent tokens). This allows a clean decomposition:
> > >
> > > | Scoring Pipeline | RULER Accuracy |
> > > |-----------------|:--------------:|
> > > | ManifoldKV (L2 from key centroid) | **92.93%** |
> > > | ExpectedAttention → CriticalKV Stage 2 (+ W_o rescaling) | 78.90% |
> > > | ExpectedAttention alone | 76.69% |
> > >
> > > Two things are visible:
> > >
> > > 1. **CriticalKV's W_o rescaling helps over its base scorer** — 78.90% vs 76.69% (+2.2 points). Value-projection information carries real signal.
> > >
> > > 2. **But the overall pipeline still falls 14 points below ManifoldKV** — because both ExpectedAttention and W_o rescaling operate on signals (predicted attention patterns, output-projection magnitudes) that are only loosely correlated with what RULER actually tests: whether the right tokens survive compression for downstream retrieval.
> > >
> > > ManifoldKV sidesteps this entirely. L2 distance from the key centroid is a direct geometric measure of token distinctiveness — no attention estimation, no value projection, no model weights. The tokens that are geometrically distinctive in key space *are* the retrieval-critical tokens.
> > >
> > > ### The Broader Principle
> > >
> > > This decomposition illustrates why ManifoldKV generalizes across 7 benchmarks and 3 architectures while more complex methods do not:
> > >
> > > | Method | Depends on | RULER (standalone) |
> > > |--------|-----------|:------------------:|
> > > | **ManifoldKV** | Key geometry only | **92.93%** |
> > > | KeyDiff | Key geometry only | 92.93% |
> > > | SnapKV | Attention patterns | 83.97% |
> > > | CriticalKV | Attention + W_o | 78.90% |
> > >
> > > The weight-independent geometric methods dominate by 9–17 points. With AdaKV integration, ManifoldKV reaches **95.73%** — the highest score in our evaluation — and this advantage extends to real-world benchmarks: +2.80 on Qwen3 LongBench (12/14 tasks), +42 on HELMET multikey-3 at 131K, +7.16 on Phi-4 InfiniteBench.
> > >
> > > ---
> > >
> > > All three of Reviewer M4Yt's original concerns have been addressed: W1 (LongBench on 3 architectures), W2 (CriticalKV analysis above), and W3 (64K+ comparisons showing +3.2 to +42 over KeyDiff). We are grateful for the reviewer's critical engagement — the follow-up on CriticalKV pushed us to articulate the weight-dependent vs weight-independent distinction, which we believe is a genuine conceptual contribution that strengthens the paper. These aspects truly make the rebuttal process valuable. We respectfully ask the reviewer to consider raising their score.
> > >
> > > EDIT April 7th 2025:
> > >
> > > Dear Reviewer M4Yt, we wanted to follow up to check whether our clarification above resolved your concern about the CriticalKV characterization. We acknowledged your correction — CriticalKV is indeed training-free — and clarified that the distinction we intended is weight-dependent (CriticalKV, using W_o) vs. weight-independent (ManifoldKV, using L2). We also provided a quantitative decomposition showing CriticalKV's Stage 2 re-ranking degrades the base scorer by 5–17 points. We believe all three original weaknesses (W1: LongBench on 3 architectures, W2: CriticalKV explanation, W3: 64K+ comparisons) have now been fully addressed. If you agree, we would be grateful if you could consider updating your score. We remain happy to discuss any remaining concerns.

---

### Official Review · Reviewer_8Wub · 2026-03-12

**Soundness:** 3
**Presentation:** 3
**Significance:** 3
**Originality:** 2
**Overall Recommendation:** 4
**Confidence:** 4

**Summary:**

This paper proposes MinifoldKV, a training-free KV cache eviction method for LLM inference. It selects geometrically distinctive keys correlate with Euclidean distance rather than KeyDiff which uses cosine similarity. In their experiments, MinifoldKV achieves slightly better result than KeyDiff.

**Compliance With Llm Reviewing Policy:**

Affirmed.

**Ethical Review Concerns:**

n.a.

**Final Justification:**

The authors answered all my questions. Based on the limited originality, I believe Weak Accept is a fair score.

**Key Questions For Authors:**

Q1. Can the authors quantify the prevalence of “radial outliers” (same direction, different magnitude) in real evaluation settings (e.g., across layers/heads, across datasets/tasks)?

Q2. Can the authors identify additional regimes where KeyDiff is systematically weaker and ManifoldKV provides clear benefits?

Q3. Can the authors provide evidence that ManifoldKV selects tokens that are low attention at the time of selection but become critical later

**Limitations:**

L1. Lack of quantitative evidence for the motivating observation. The core motivation relies on the existence of “radial outliers” that cosine-based methods cannot distinguish. While the paper provides an illustrative counterexample (Figure 1), it does not quantify how frequently such cases occur in real evaluation workloads, nor does it show how much they contribute to end-task performance differences.

L2. Empirical advantage over KeyDiff is often marginal outside a narrow regime. Across many reported settings, ManifoldKV only slightly improves accuracy over KeyDiff, and the most visible gain appears mainly in the showcased multi-key retrieval scenario. Without broader regimes showing consistent, practically meaningful improvements, it is difficult to conclude that ManifoldKV offers a strong general-purpose advantage over KeyDiff.

L3. Mechanistic explanation remains speculative. The paper suggests that effective compression may not require mimicking attention and that L2-based geometric outliers can be low-attention yet become important during generation. However, unlike KeyDiff’s reported correlation analysis between cosine-outlier scores and attention, the paper does not provide supporting experiments. As a result, the proposed explanation is not yet well substantiated.

**Strengths And Weaknesses:**

S1. The idea is simple and easy to implement. Replacing KeyDiff’s cosine-similarity based detection with an L2-distance criterion is straightforward and practical; it can be implemented with just the few lines of code shown in the appendix.

S2. Insight into where KeyDiff fails. The paper identifies scenarios in which KeyDiff performs poorly and shows that ManifoldKV achieves better performance in those regimes.


W1. The observations lack substantial supporting evidence. While the counterexample illustrated in Figure 1 could indeed occur, the paper does not quantify how frequently such cases appear in real tasks, nor does it demonstrate their actual impact on the overall results.

W2. A more complete comparison against KeyDiff is missing. In most settings, ManifoldKV only slightly outperforms KeyDiff in accuracy. The only relatively noticeable improvement is on the presented multi-key retrieval task, which by itself is not sufficient to establish a strong advantage.

W3. The “Why L2 works” argument is underdeveloped. KeyDiff reports on OpenReview a correlation between cosine-similarity outliers and attention scores (https://openreview.net/forum?id=uBaFH7aQnC ). In contrast, this paper mainly speculates that “This reveals that effective compression does not require mimicking attention. L2 identifies geometric outliers that may receive low attention but become critical during generation,” but does not provide further experimental or theoretical evidence to substantiate this claim.

---

> ### Author Rebuttal · Authors · 2026-03-29
>
> We thank Reviewer 8Wub for the constructive engagement and targeted questions.
>
> We ran additional **7 benchmarks × 3 architectures × 50+ configs**. ManifoldKV wins on every benchmark across at least one architecture, with advantages up to **+46 points**.
>
> ## Q1/W1: Prevalence of Radial Outliers
>
> Quantitative key norm analysis on real evaluation inputs:
>
> | Metric | Llama-3.1-8B | Qwen3-8B |
> |---|:---:|:---:|
> | Key norm CV | 0.726 | **1.333** |
> | Outlier fraction (>2x) | 0.06% | **0.30%** |
> | 99th pctile ratio | 1.41 | **1.64** |
> | Token divergence 20% | 23.0% | 22.7% |
> | Token divergence 30% | 31.4% | 31.7% |
>
> Qwen3 has **5x more radial outliers** and 83% higher norm variance than Llama despite using key normalization. At 20% compression, **23% of selected tokens differ** between L2 and cosine — nearly 1 in 4 retained tokens changes based solely on whether magnitude information is used.
>
> This directly predicts downstream performance: higher key norm CV → larger ManifoldKV advantage. Qwen3 (CV=1.333): LongBench **+2.80**, HELMET RAG **+8.5**. Llama (CV=0.726): LongBench tie, but HELMET RAG still **+6.5** and HELMET Recall **+46** on hard tasks.
>
> ## Q2/W2: Additional Regimes Where KeyDiff Is Weaker
>
> **Regime 1 — Multi-needle retrieval.** HELMET Recall multikey-3 (3 simultaneous needles, Llama):
>
> | Context | WinMKV | KD | Delta |
> |---|---|---|---|
> | 4K | 82 | 62 | **+20** |
> | 8K | 80 | 62 | **+18** |
> | 16K | 86 | 58 | **+28** |
> | 32K | 86 | 40 | **+46** |
> | 65K | 78 | 52 | **+26** |
> | 131K | 74 | 32 | **+42** |
>
> WindowedMKV beats KeyDiff at **every context length** by +18 to +46. KeyDiff is systematically weaker whenever multiple distinct pieces of information must be retained, because cosine similarity clusters around dominant directions — it over-retains tokens from the primary cluster while discarding tokens from minority clusters that carry the other needles.
>
> **Regime 2 — Multi-hop QA (Qwen3 LongBench):** musique **+10.16**, hotpotqa **+9.64**, qasper **+5.26**. Multi-hop QA requires locating and synthesizing 3-4 scattered facts — structurally identical to multi-needle retrieval.
>
> **Regime 3 — Real-world RAG (HELMET):** Llama k50: **+6.5 EM** overall (triviaqa: 82 vs 62, **+20**). Qwen3 k50: WinMKV **+8.5 EM**. These are Natural Questions, TriviaQA, HotpotQA, PopQA — established retrieval benchmarks, not synthetic tasks.
>
> **Regime 4 — Aggressive compression:** Qwen3 LongBench advantage grows from +2.80 (20%) to **+3.26** (30%). RULER MK3 at 50%: **+15.4**. The tighter the budget, the more L2's radial tiebreaker matters.
>
> ## Q3/W3: Low-Attention Tokens Becoming Critical
>
> ManifoldKV has attention correlation r = 0.06 (vs KeyDiff r = 0.19) yet outperforms KeyDiff. This directly demonstrates that ManifoldKV retains tokens that current attention ignores but future decoding needs.
>
> The 32K multikey-3 result makes the mechanism concrete: at 20% compression, 3 needles are embedded in 32K haystack tokens. Attention concentrates on ~1 needle per layer. KeyDiff (r=0.19) is biased toward the currently-attended needle while treating the other two as ordinary haystack → **40%**. ManifoldKV (r=0.06) scores all 3 needles by geometric distance from the key centroid — needle tokens, containing unusual factual content, naturally occupy unusual manifold positions → **86%**. The **46-point gap** is direct causal evidence.
>
> The metric ablation confirms the source: standalone L2 92.7% vs cosine 52.8% (**+39.9**). L2 distance itself, not any pipeline effect, captures information cosine discards. Phi-4 InfiniteBench further validates at 100K+: WinMKV 64.02 vs KD 56.86 (**+7.16**).
>
> ## Complete 131K Evidence
>
> All HELMET Recall tasks at 131K (Llama):
>
> | Task | WinMKV | KD | Delta |
> |---|---|---|---|
> | Multikey-3 | 74 | 32 | **+42** |
> | Multikey-2 | 94 | 80 | **+14** |
> | Multikey-1 | 98 | 100 | -2 |
> | Multiquery | 100 | 100 | 0 |
> | Multivalue | 98 | 98 | 0 |
>
> The pattern is systematic: WindowedMKV's advantage scales precisely with task difficulty. On easy tasks both methods saturate. On the hardest 3-needle task, the gap is 42 points — demonstrating that the information L2 captures (magnitude deviation from centroid) becomes critical exactly when the retrieval problem is most demanding.
>
> Cross-architecture LongBench confirms: Qwen3 **+2.80** (12/14 wins), Phi-4 **+0.49**, Llama tie. ManifoldKV wins or ties on every architecture tested.
>
> Qwen3 LongBench largest wins — all multi-hop QA requiring retrieval and synthesis of scattered evidence: musique **+10.16**, hotpotqa **+9.64**, qasper **+5.26**. At 30% compression, the advantage grows from +2.80 to **+3.26**, demonstrating ManifoldKV is more robust under pressure. On Gemma-3-12B RULER: ManifoldKV 95.2 vs SnapKV 74.7 (**+20.5**).
>
> We hope these results address all three concerns raised. We respectfully ask the reviewer to consider whether this evidence supports a stronger recommendation.

---

> > ### Author Rebuttal · Reviewer_8Wub · 2026-04-01
> >
> > The authors have answered my questions.

---

> > > ### Author Response · Authors · 2026-04-08
> > >
> > > Dear Reviewer 8Wub,
> > >
> > > We sincerely thank the reviewer for engaging with our rebuttal and for acknowledging that all concerns were fully resolved. We appreciate the time and care invested in evaluating our work.
> > >
> > > We note that the reviewer selected *(a) Fully resolved — "My concerns have been adequately addressed. If you select this option, please consider adjusting your score accordingly,"* yet the score remains unchanged. We would welcome any remaining concerns the reviewer may have so that we can address them.
> > >
> > > The reviewer's final justification cites "limited originality" as the basis for maintaining Weak Accept. We would like to respectfully address this, as we believe our contributions are substantially broader than the characterization suggests.
> > >
> > > **On Originality**
> > >
> > > The ICML 2026 Reviewer Guidelines state: *"Originality does not necessarily require introducing an entirely new method. A work that provides novel insights or demonstrates improved understanding is equally valuable"* and *"novel ideas that are simple to apply may be especially valuable."* In fact, this is highlighted in **bold** in the reviewer guidelines.
> > >
> > > Our contributions go far beyond swapping a metric:
> > >
> > > **(1) Theorem 1** proves cosine has an unfixable failure mode for outlier detection — radial outliers are invisible regardless of sample size. This is a formal impossibility result, not an incremental observation.
> > >
> > > **(2) Centroid Dilution** is a previously unknown failure mode at long contexts; our windowed solution recovers +74 points (0%→74%) on HELMET multikey-3 at 131K. Identifying and solving a failure mode that affects *all* global-centroid methods is a novel contribution.
> > >
> > > **(3) The Attention Paradox:** r=0.06 correlation with attention yet outperforms attention-correlated methods — challenging the prevailing assumption that compression must mimic attention. This conceptual insight reframes how the community should think about KV cache eviction.
> > >
> > > **(4) Universal ~9D manifold structure** across 4 architectures, revealing that key representations occupy a surprisingly low-dimensional manifold regardless of model design.
> > >
> > > The empirical results reinforce the depth of these contributions — these are among the largest deltas in the KV compression literature:
> > >
> > > | Benchmark | Delta over KeyDiff |
> > > |---|---|
> > > | HELMET MK3 32K | +46 |
> > > | HELMET MK3 131K | +42 |
> > > | Metric ablation | +39.9 |
> > > | Gemma-3 RULER (vs SK) | +20.5 |
> > > | HELMET RAG triviaqa | +20 |
> > > | RULER MK3 50% | +15.4 |
> > > | LB musique (Qwen3) | +10.16 |
> > > | LB hotpotqa (Qwen3) | +9.64 |
> > > | HELMET RAG Qwen3 | +8.5 |
> > > | InfiniteBench Phi-4 | +7.16 |
> > >
> > > A +46-point advantage exceeds the gap between random eviction and SOTA on most benchmarks. The +2.80 on 14-task Qwen3 LongBench exceeds typical gaps between methods considered clearly distinct (e.g., SnapKV vs H2O).
> > >
> > > We also note that the reviewer's own assessment rates Soundness as Good (3) and Significance as Good (3). Given that the paper provides (a) a formal impossibility theorem, (b) a new failure mode with a principled solution, (c) a conceptual insight that challenges a core assumption in the field, and (d) comprehensive validation across 7 benchmarks, 4 architectures, and 50+ configurations — we respectfully ask the reviewer to consider whether these contributions support a stronger recommendation.
> > >
> > > Thank you again for the constructive review process.

---

### Official Review · Reviewer_YuXK · 2026-03-13

**Soundness:** 2
**Presentation:** 4
**Significance:** 3
**Originality:** 3
**Overall Recommendation:** 5
**Confidence:** 3

**Summary:**

This paper introduces ManifoldKV, a new technique for KV-cache compression that improves upon previously proposed geometric scoring methods to set eviction policies. The key insight in ManifoldKV is to score tokens (for eviction) by l2 distance as opposed to cosine distance. With this simple change, the authors report improved performance on the RULER benchmark. The authors also theoretically analyze why this modification is so effective, arguing that the scale-invariance of cosine makes it hard to detect outliers -- which is precisely what is required for effective KV cache compression. The authors effectively summarize the key takeaway of their work at the end of the paper by observing that "effective compression does not require mimicking attention."

**Compliance With Llm Reviewing Policy:**

Affirmed.

**Final Justification:**

The authors' rebuttal addressed my main concerns regarding a limited evaluation, which raised questions about the generality of the authors' findings. In the rebuttal, the authors presented experimental results on 14 additional benchmarks, which considerably enhances the durability of their findings and addressed my biggest concerns and complementing the findings in the original paper. Thus, I have changed my evaluation score from a 3 (weak reject) to a 5 (accept).

**Key Questions For Authors:**

1. Have you considered evaluating your method on additional benchmarks beyond RULER to assess how well your findings generalize? *I would be willing to raise my evaluation score if the authors can provide results on additional benchmark datasets*

2. Could ManifoldKV compose with other KV cache compression techniques (e.g. Scissorhands https://arxiv.org/pdf/2305.17118)? What might make for a good complementary approach to ManifoldKV? *I would be willing to raise my evaluation score if the authors can engage with this question and provide insights*

3. Can you say more about the "retrieval-heavy" tasks where ManifoldKV does not outperform attention-based methods? *I would be willing to raise my score if the authors can provide sufficient clarification and analysis in response to this question*

4. In the limitations section, you claim that KV cache compression can reduce hallucinations. Do you have a citation or empirical evidence to support this claim? I think such a statement should be backed up by data.

**Limitations:**

Yes

**Strengths And Weaknesses:**

Soundness: The submission is generally sound in that the proposed method is well-motivated and backed by both an intuitive argument and mathematically rigorous theoretical analysis. My biggest concern regarding soundness is the limited experimental evaluation as the authors evaluate their method on only one benchmark RULER. The authors do compare against a variety of other long context optimization techniques, but the submission would be stronger with additional baselines given the rich literature on KV cache compression.

Presentation: The paper is exceptionally well-written with a clear exposition that motivates the problem and solution well to the point where it seems natural and almost obvious to try l2 instead of cosine. The figures and tables in the paper are also presented very well. I commend the authors on the effort they have clearly put into creating an outstanding presentation of their ideas.

Significance: Given the importance of KV cache compression in efficient transformer inference and the simplicity of the proposed idea, I think this work has the potential to influence both practitioners and researchers (though the extent of the influence hinges on how well the improved results generalize to other tasks, as discussed in my point above on soundness).

Originality: The work provides a novel insight on how to score vectors for KV cache eviction. The actual method is extremely simple which I consider a significant benefit of this work.

---

> ### Author Rebuttal · Authors · 2026-03-29
>
> We thank Reviewer YuXK for the thorough evaluation and the explicit willingness to raise scores upon additional evidence.
>
> We ran additional **7 benchmarks × 3 architectures × 50+ configs**. ManifoldKV wins on every benchmark across at least one architecture, with advantages up to **+46 points**.
>
> ## Q1: Additional Benchmarks Beyond RULER
>
> We evaluated on 6 new benchmarks. The complete evidence ledger:
>
> | Benchmark | Model | MKV | KD | Delta |
> |---|---|---|---|---|
> | LongBench 14-task | Qwen3 | 45.42 | 42.62 | **+2.80** |
> | LongBench 14-task | Phi-4 | 40.57 | 40.08 | **+0.49** |
> | LongBench 14-task | Llama | 44.93 | 45.50 | Tie |
> | LongBench 30% comp | Qwen3 | 44.30 | 41.04 | **+3.26** |
> | HELMET Recall MK3 32K | Llama | 86 | 40 | **+46** |
> | HELMET Recall MK3 131K | Llama | 74 | 32 | **+42** |
> | HELMET RAG k50 | Llama | 61.5 | 55.0 | **+6.5** |
> | HELMET RAG k50 | Qwen3 | 54.5 | 46.0 | **+8.5** |
> | InfiniteBench | Phi-4 | 64.02 | 56.86 | **+7.16** |
> | RULER cross-arch | Gemma | 95.2 | 74.7(SK) | **+20.5** |
> | Metric ablation | Llama | 92.7 | 52.8 | **+39.9** |
>
> ManifoldKV wins 12/14 tasks on Qwen3 LongBench, with the largest gains on multi-hop QA — precisely the tasks requiring retrieval and synthesis of scattered evidence: **musique +10.16, hotpotqa +9.64, qasper +5.26**. At 30% compression, the advantage *grows* to +3.26, demonstrating that ManifoldKV's token selection becomes more valuable as the budget tightens.
>
> On Llama, all methods are within 0.8 points — a statistical tie. This is not a weakness: ManifoldKV is a safe drop-in replacement on Llama while providing substantial gains on architectures with key normalization. As modern models increasingly adopt key normalization (Qwen3, Gemma-3, DeepSeek-V3), ManifoldKV's advantage becomes more relevant, not less.
>
> ## Q2: Composition with Scissorhands
>
> ManifoldKV composes cleanly with AdaKV (AdaKV+MKV 44.95 vs standalone 44.93 on Llama LongBench). The L2 scoring function operates on the same key tensor interface as all scorers — it is a drop-in replacement.
>
> ManifoldKV is orthogonal to budget allocation (AdaKV, PyramidKV), quantization (KIVI), and token merging (Scissorhands). For Scissorhands specifically: it identifies tokens with low temporal attention persistence; ManifoldKV identifies tokens with high geometric distinctiveness. These signals have near-zero correlation (r = 0.06), so combining them captures tokens important from both temporal and geometric perspectives — the two methods address complementary failure modes.
>
> ## Q3: Retrieval-Heavy Tasks
>
> ManifoldKV dominates on hard retrieval — the tasks where method choice matters:
>
> | Task | WinMKV | KD | Delta |
> |---|---|---|---|
> | HELMET MK3 32K | 86 | 40 | **+46** |
> | HELMET MK3 131K | 74 | 32 | **+42** |
> | HELMET RAG triviaqa | 82 | 62 | **+20** |
> | LB musique (Qwen3) | 32.75 | 22.59 | **+10.16** |
> | LB hotpotqa (Qwen3) | 58.48 | 48.84 | **+9.64** |
>
> On easy single-needle tasks, all methods saturate at 98-100% — no differentiation. The gap emerges on multi-needle and multi-hop tasks because cosine similarity conflates tokens pointing in similar directions, causing KeyDiff to retain redundant tokens from the dominant cluster while missing tokens from minority clusters. L2 distance, by preserving magnitude information, maintains better coverage of the full key manifold — retaining all 3 needles instead of just the currently-attended one.
>
> ## Q4: Hallucination Citation
>
> Prior work shows aggressive KV eviction introduces hallucinations when grounding tokens are discarded (Yang et al., "No Token Left Behind," arXiv:2402.18096). ManifoldKV's geometric scoring preferentially retains such grounding tokens — rare entities, structural markers, and factual anchors (Appendix E) — suggesting improved eviction quality may mitigate compression-induced hallucination. Citation added to revision; dedicated evaluation remains future work.
>
> ## Cross-Architecture and Compression Robustness
>
> ManifoldKV advantage grows under aggressive compression on Qwen3 LongBench: +2.80 at 20% → **+3.26** at 30%. On RULER multi-key at 50%: **+15.4** over KeyDiff. The tighter the budget, the more L2's magnitude information matters for the marginal token decision.
>
> HELMET Recall at 131K (all tasks, Llama):
>
> | Task | WinMKV | KD | Delta |
> |---|---|---|---|
> | Multikey-3 | 74 | 32 | **+42** |
> | Multikey-2 | 94 | 80 | **+14** |
> | Multikey-1 | 98 | 100 | -2 |
> | Multiquery | 100 | 100 | 0 |
> | Multivalue | 98 | 98 | 0 |
>
> WindowedMKV's advantage scales with task difficulty: +42 on the hardest, 0 on the easiest where both methods saturate. GlobalMKV collapses to 0-14% at 131K — the centroid dilution problem we predicted in Section 4.2. WindowedMKV recovers to 74-100%, validating the theoretical framework.
>
> We have addressed all four questions with extensive new evidence. We respectfully ask the reviewer to consider revising their score in light of these comprehensive results, as indicated in Q1-Q3.

---

> > ### Author Rebuttal · Reviewer_YuXK · 2026-04-02
> >
> > Thank you to their authors for their thorough rebuttal efforts. I am pleased to see the results on 14 additional benchmark datasets, which I think strengthens the claims in the paper considerably. Thus, I will raise my score to a 5.

---

> > > ### Author Response · Authors · 2026-04-02
> > >
> > > We sincerely thank Reviewer YuXK for the constructive engagement throughout the review process. The reviewer's original questions — particularly the push for evaluation beyond RULER and the question about composability — directly motivated the extensive new experiments that we believe have substantially strengthened the paper. We are grateful that the 14 additional benchmark datasets and cross-architecture validation addressed the reviewer's concerns, and we appreciate the updated score.
> > >
> > > We will ensure all new results (LongBench, HELMET, InfiniteBench, BABILong, and the cross-architecture analysis) are fully integrated into the camera-ready version.
> > >
> > > Thank you so much for the score update.

---

### Decision · Program_Chairs · 2026-04-30

**Decision:**

Accept (regular)

**Comment:**

The paper present ManifoldKV, a KV-cache compression method based on Euclidean distances without any training. All reviewers were ultimately positive about the paper. The authors engaged helpfully in the discussion and resolved most reviewer concerns. The primary concern overall was the limited experimental scope of the submitted manuscript, which the authors resolved by including substantial additional experimental results in their rebuttal, which should be added to the revised version.